# PROTO: Iterative Policy Regularized Offline-to-Online Reinforcement Learning

## Abstract

Offline-to-online reinforcement learning (RL), by combining the benefits of offline pretraining and online finetuning, promises enhanced sample efficiency and policy performance. However, existing methods, effective as they are, suffer from suboptimal performance, limited adaptability, and unsatisfactory computational efficiency. We propose a novel framework, *PROTO*, which overcomes the aforementioned limitations by augmenting the standard RL objective with an iteratively evolving regularization term. Performing a trust-region-style update, *PROTO* yields stable initial finetuning and optimal final performance by gradually evolving the regularization term to relax the constraint strength. By adjusting only a few lines of code, *PROTO* can bridge any offline policy pretraining and standard off-policy RL finetuning to form a powerful offline-to-online RL pathway, birthing great adaptability to diverse methods. Simple yet elegant, *PROTO* imposes minimal additional computation and enables highly efficient online finetuning. Extensive experiments demonstrate that *PROTO* achieves superior performance over SOTA baselines, offering an adaptable and efficient offline-to-online RL framework.

## 1 Introduction

Reinforcement learning (RL) holds the potential to surpass human-level performances by solving complex tasks autonomously (Silver et al., 2017). However, collecting a large amount of online data, especially the initial random explorations, can be expensive or even hazardous (Nair et al., 2020). Offline RL and offline imitation learning (IL) offer alternatives to training policies without environment interactions, by exploiting fixed offline datasets generated by a behavior policy. However, their performances are heavily limited by the quality and state-action space coverage of pre-existing offline datasets (Jin et al., 2021). This largely inhibits these approaches in real-world applications, where both sample efficiency and optimal performance are required (Kormushev et al., 2013).

Offline-to-online RL (Nair et al., 2020) has emerged as a promising solution, by pretraining a policy $\pi_0$ using offline RL/IL and then finetuning with online RL. Ideally, offline-to-online RL can improve sample efficiency with favorable initialization for online RL. Further, by exploring more high-quality data, it overcomes the suboptimality of offline RL/IL caused by the over-restriction on a fixed suboptimal dataset. However, directly finetuning a pretrained policy often suffers from severe (even non-recoverable) performance drop at the initial finetuning stage, caused by distributional shift and the over-estimation error of value function at out-of-distribution (OOD) regions (Nair et al., 2020).

Existing works typically adopt conservative learning to alleviate this initial performance drop, which has three major drawbacks: 1) *Suboptimal performance.* The majority of current methods introduce policy constraints to combat performance drop due to distributional shift (Nair et al., 2020). Optimizing policy with an additional constraint, however, impedes the online learning process and can cause non-eliminable suboptimality gap (Kumar et al., 2019). 2) *Limited adaptability.* Most existing methods are tailored to a specific pretraining or finetuning method, which lacks adaptability to bridge diverse methods to achieve the best possible performance (Lee et al., 2022). 3) *Computational inefficiency.* Moreover, some works require ensemble models to obtain near-optimal performance (Lee et al., 2022), which introduces tremendous computational costs and is unscalable to larger models.

In this paper, we propose a generic and adaptive framework, *iterative Policy Regularized Offline-To-Online RL (PROTO)*, which incorporates an iterative policy regularization term into the standard RL

objective. Performing a trust-region-style update (Schulman et al., 2015), our method encourages the finetuning policy to remain close to $\pi_k$ (policy at last iteration, with $\pi_0$ being the pretrained policy). Compared to existing methods, *PROTO* adopts appropriate conservatism to overcome the initial performance drop, while gradually relaxing excessive restrictions by casting constraint on an evolved $\pi_k$ rather than the fixed $\pi_0$, which leads to stable and optimal finetuning performance. Moreover, theoretical analysis proves that the introduced iterative regularization term induces no suboptimality and hence is far more *optimistic* compared to previous policy constraints that typically cause suboptimal performance due to over-conservatism. Therefore, *PROTO* recognizes the necessity of giving enough freedom to finetuning in order to obtain near-optimal policies. It imposes minimal assumptions on pretraining and finetuning methods, allowing for seamless extension to diverse methods accomplished by adding just a few lines of code to standard off-policy RL finetuning. Simple yet effective, *PROTO* achieves state-of-the-art performance on D4RL benchmarks (Fu et al., 2020) and introduces negligible computational costs, retaining high computational efficiency on par with standard off-policy RL approaches and offering a competitive offline-to-online RL framework.

## 2 RELATED WORK

**Policy Constraint (PC).** The most straightforward way to mitigate the initial finetuning performance drop is to introduce policy constraints to combat the distributional shift. Existing methods, however, are over-conservative as they typically constrain the policy in a fixed constraint set (e.g., offline dataset support (Kumar et al., 2019)), which can lead to severely suboptimal performance (Kumar et al., 2019). Nair et al. (2020) is the first offline-to-online RL approach that obtains stable finetuning performance. It introduces advantage weighted regression (AWR) (Peng et al., 2019) to extract policy, which is equivalent to implicitly constraining the policy *w.r.t.* the replay buffer $\mathcal{B}$ that is updated by filling in newly explored transitions. Some offline RL approaches adopt AWR-style policy extraction to learn policies that can be directly utilized for online finetuning (Kostrikov et al., 2022; Garg et al., 2023; Xiao et al., 2023). AWR, however, cannot be plugged into diverse online RL approaches non-intrusively, limiting its adaptability. Sharing similar philosophy, some works constrain the policy to stabilize training, but using a pluggable regularization (Wu et al., 2022; Zhao et al., 2022; Zheng et al., 2023) such as simply adding one additional IL loss (Wu et al., 2022), which is easily adaptable to diverse online finetuning approaches. All these methods are over-conservative since the constraint on a mixed replay buffer $\mathcal{B}$ or a behavior policy $\mu$ may be severely suboptimal (Kumar et al., 2019; Li et al., 2023; Wu et al., 2022). Some recent works partially reduce the over-conservatism by constraining on a potentially well-performing pretrained policy $\pi_0$ (Yu & Zhang, 2023; Agarwal et al., 2022; Zhang et al., 2023). However, $\pi_0$ may still be severely suboptimal when pretrained on a suboptimal offline dataset (Kumar et al., 2019; Jin et al., 2021).

**Pessimistic Value Initialization (PVI).** One alternative to address performance drop is to initialize online RL with a pessimistic value function, to alleviate the side effect of overestimation errors. By doing so, the value function already attains low values at OOD regions and one can directly finetune online RL without introducing any conservatism, which has the potential to obtain near-optimal finetuning performance. Lee et al. (2022) is the first to adopt pessimistic value initialization and introduces a balanced experience replay scheme. Nakamoto et al. (2023) further improves upon (Lee et al., 2022) by conducting a simple value surgery to ameliorate training instability caused by the over-conservative value initialization at OOD regions. However, these methods heavily rely on CQL (Kumar et al., 2020) pretraining framework, which inherit the main drawbacks of CQL such as being over-conservative and computationally inefficient (Kostrikov et al., 2021; Li et al., 2023). Thus, when tasks are too difficult for CQL to obtain reasonable initialization, inferior performance may occur. Moreover, an ensemble of pessimistic value functions is generally required to better depict the manifold of OOD regions (Lee et al., 2022), which again inevitably imposes tremendous computational costs during both offline pretraining and online finetuning.

**Goal-Conditioned Supervised Learning (GCSL).** A recent study (Zheng et al., 2022) considers the decision transformer (DT) (Chen et al., 2021) finetuning setting and introduces entropy regularization to improve exploration. However, DT is formulated as a conditioned-supervised learning problem, which can be perceived as implicitly constraining policies on the replay buffer $\mathcal{B}$ similar to AWR, hence also suffering suboptimal performance when $\mathcal{B}$ is severely suboptimal.

Table 1: Comparison of existing practical offline-to-online RL methods. See Table 2 in Appendix A for a more detailed comparison of other offline-to-online RL methods. $\mu$: behavior policy that generates the offline dataset $\mathcal{D}$. $\mathcal{B}$: replay buffer. $\pi_0$: pretrained policy. SPOT (Wu et al., 2022), AWAC (Nair et al., 2020), IQL (Kostrikov et al., 2022), PEX (Zhang et al., 2023), Off2On (Lee et al., 2022), ODT (Zheng et al., 2022).

| Type | | PC | | | PVI | GCSL |
|---|---|---|---|---|---|---|
| Method | SPOT | AWAC | IQL | PEX | Off2On | ODT |
| a. Constraint policy set | $\mu$ | | $\mathcal{B}$ | $\pi_0$ | No Constraint | $\mathcal{B}$ |
| b. Stable and optimal policy learning | ✗ | | ✗ | ✗ | ✓ | ✗ |
| c. Adaptable to diverse pretraining methods | ✓ | ✓ | | ✓ | ✗ | ✗ |
| d. Adaptable to diverse finetuning methods | ✓ | ✗ | | ✓ | ✓ | ✗ |
| e. Computationally efficient | ✓ | ✓ | | ✓ | ✗ | ✗ |

## 3 PROTO RL FRAMEWORK

### 3.1 PROBLEM DEFINITION

We consider the infinite-horizon Markov Decision Process (MDP) (Puterman, 2014), which is represented by a tuple $\mathcal{M} := \langle \mathcal{S}, \mathcal{A}, r, \rho, \mathcal{P}, \gamma \rangle$, where $\mathcal{S}$ and $\mathcal{A}$ denote the state and action space, respectively. $r : \mathcal{S} \times \mathcal{A} \to \mathbb{R}$ represents a reward function, $\rho$ denotes initial distribution, $\mathcal{P} : \mathcal{S} \times \mathcal{A} \to \mathcal{S}$ is the transition kernel, and $\gamma \in (0, 1)$ is a discount factor.

Standard RL aims to learn a policy $\pi^* : \mathcal{S} \to \mathcal{A}$ that maximizes the expected discounted return $J(\pi) = \mathbb{E}\left[\sum_{t=0}^{\infty} \gamma^t r(s_t, a_t) | s_0 \sim \rho, a_t \sim \pi(\cdot|s_t), s_{t+1} \sim \mathcal{P}(\cdot|s_t, a_t)\right]$, i.e., $\pi^* \leftarrow \arg\max_\pi J(\pi)$. One popular approach to solving the above problem is approximate dynamic programming (ADP) (Powell, 2007), which typically approximates the action-value function $Q^{\pi_k}(s, a)$ of the policy $\pi_k$ at the last iteration by repeatedly applying the following policy evaluation operator $\mathcal{T}^{\pi_k}, k \in N$:

$$(\mathcal{T}^{\pi_k} Q)(s, a) := r(s, a) + \gamma \mathbb{E}_{s' \sim \mathcal{P}(\cdot|s,a), a' \sim \pi_k(\cdot|s')} [Q(s', a')] \tag{1}$$

Then, standard actor-critic RL approaches introduce one additional policy improvement step to further optimize the action-value function $Q^{\pi_k}(s, a)$ (Lillicrap et al., 2016; Haarnoja et al., 2018):

$$\pi_{k+1} \leftarrow \arg\max_\pi \mathbb{E}_{a \sim \pi(\cdot|s)}[Q^{\pi_k}(s, a)] \tag{2}$$

In high-dimensional or continuous space, $Q^{\pi_k}$ is generally learned by enforcing the single-step Bellman consistency, i.e., $\min_Q J^{\pi_k}(Q) = \frac{1}{2}\mathbb{E}_{(s,a,s') \sim \mathcal{B}}\left[\left((\mathcal{T}^{\pi_k} Q - Q)\right)(s, a)\right]^2$, where $\mathcal{B}$ is a replay buffer that is updated by filling in new transitions during the training process. The policy improvement step is also performed on this replay buffer, i.e., $\pi_{k+1} \leftarrow \arg\max_\pi \mathbb{E}_{s \sim \mathcal{B}, a \sim \pi(\cdot|s)} [Q^{\pi_k}(s, a)]$.

### 3.2 OFFLINE-TO-ONLINE RL

Offline-to-online RL ensures favorable initialization for online RL with a pretrained policy, meanwhile overcomes the suboptimality of offline RL or IL by exploring more high-quality data with online finetuning. However, directly finetuning offline pretrained policy with online RL often suffers from severe performance drop caused by distributional shift and over-estimation error at OOD regions (Nair et al., 2020). Thus, additional regularization is required to stabilize the finetuning process. Since optimizing policy with additional regularization can lead to suboptimal performance (Kumar et al., 2019; Li et al., 2023), the primary goal of the offline-to-online RL pathway is to balance stability and optimality during online finetuning. This requires policy finetuning to be initially stable while avoiding excessive conservatism to achieve near-optimal policies.

**Limitations of SOTA**. As summarized in Table 1, previous offline-to-online RL studies all directly borrow conservatism from offline RL to stabilize online finetuning. Current methods, especially those based on policy constraint and goal-conditioned supervised learning, prioritize stability over policy performance optimality (Nair et al., 2020; Kostrikov et al., 2022; Zheng et al., 2022), by keeping policy constraints fixed (e.g., behavior policy $\mu$ and pretrained policy $\pi_0$) or changed slowly during online finetuning (e.g., mixed replay buffer $\mathcal{B}$). Thus, if the initial constraints are severely suboptimal, they may restrain the online finetuning process to *suboptimal performance with poor online sample efficiency*, as illustrated in Figure 1.

Some works gradually anneal the constraint strength to alleviate over-conservatism (Wu et al., 2022; Agarwal et al., 2022). However, even with constraint annealing, suboptimal and slow online finetuning still occurs if the initial over-conservatism is too strong, as shown in Figure 1. Therefore, directly using fixed policy constraints may not be the best choice for offline-to-online RL. Recent pessimistic value initialization method provides stable and optimal policy learning without introducing additional conservatism into online finetuning (Lee et al., 2022; Nakamoto et al., 2023; Hong et al., 2023), but at the expense of inefficiency, as it requires ensemble models to achieve reasonable performance with significant computational overhead. So far, it still remains a challenge how to strike a *balance between stability and optimality* in a *computationally efficient* way.

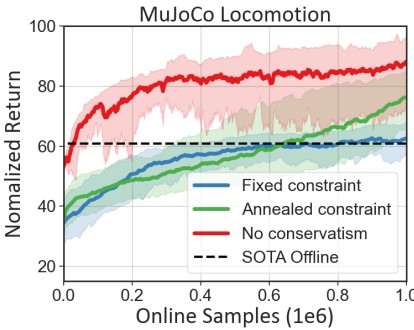

Figure 1: Aggregated learning curves of online finetuning with different policy constraints on 9 MuJoCo Locomotion tasks in D4RL benchmark (Fu et al., 2020). When policy constraints are involved, severely suboptimal performance persists. Fixed constraint: IQL (Kostrikov et al., 2022); Annealed constraint: Frozen (see Section 4.4); No conservatism: Off2On (Lee et al., 2022).

In addition, most previous studies focus on a specific pretraining or finetuning method (Nair et al., 2020; Kostrikov et al., 2022; Lee et al., 2022), with *limited adaptability* to diverse RL approaches. An ideal offline-to-online RL, however, should provide a universal solution that bridges a wide range of offline pretraining and online finetuning approaches to achieve the best possible performance and applicability. The simplest way to achieve this is to adopt a pluggable regularization, such as adding a BC term into the original policy loss (Fujimoto & Gu, 2021; Wu et al., 2022), which can be easily integrated into diverse RL methods. However, such regularization often suffers from large suboptimality gaps (Wu et al., 2022) due to lack of consideration on policy optimality.

In summary, conservative designs commonly used in offline RL, such as policy constraints, pessimistic value regularization, and goal-conditioned supervised learning, inevitably suffer from suboptimal performance, limited adaptability, and inefficiency. Next, we introduce a new easy-to-use regularization that effectively retains stability while enabling adaptive and efficient online policy learning.

### 3.3 ITERATIVE POLICY REGULARIZATION

Trust-region update has gained some success in online RL (Schulman et al., 2015; 2017; Nachum et al., 2018) and thanks to its potential for unifying offline and online policy learning, has recently been extended to solve offline RL problems (Zhuang et al., 2023). Inspired by this, we propose a generic and adaptive framework, *iterative Policy Regularized Offline-To-Online RL (PROTO)*, which augments the standard RL objective $J(\pi)$ with an *Iterative Policy Regularization* term:

$$\pi_{k+1} \leftarrow \arg\max_{\pi} \mathbb{E} \left[ \sum_{t=0}^{\infty} \gamma^t \left( r(s_t, a_t) - \alpha \cdot \log \left( \frac{\pi(a_t|s_t)}{\pi_k(a_t|s_t)} \right) \right) \right], k \in N, \quad (3)$$

where $\pi_k$ is the policy at the last iteration, with $\pi_0$ being the pretrained policy. This objective seeks to simultaneously maximize the reward and minimize the KL-divergence *w.r.t.* the policy obtained at the last iteration $\pi_k$, which is equivalent to optimizing the original objective within the log-barrier of $\pi_k$, hence can be interpreted as a trust-region-style learning objective.

Similar to the treatment in Max-Entropy RL (Haarnoja et al., 2018), this *Iterative Policy Regularized* MDP gives the following policy evaluation operator by simply adding a regularization term into Eq. (1):

$$(\mathcal{T}_{\pi_{k-1}}^{\pi_k} Q)(s, a) := r(s, a) + \gamma \mathbb{E}_{s' \sim \mathcal{P}(\cdot|s,a), a' \sim \pi_k(\cdot|s')} \left[ Q(s', a') - \alpha \cdot \log \left( \frac{\pi_k(a'|s')}{\pi_{k-1}(a'|s')} \right) \right], k \in N^+, \quad (4)$$

The policy improvement step can be realized by adding a similar regularization term into Eq. (2):

$$\pi_{k+1} \leftarrow \arg\max_{\pi} \mathbb{E}_{a \sim \pi(\cdot|s)} \left[ Q^{\pi_k}(s, a) - \alpha \cdot \log \left( \frac{\pi(a|s)}{\pi_k(a|s)} \right) \right], k \in N. \quad (5)$$

Despite its simplicity, we will show that *PROTO* can naturally balance the stability and optimality of policy finetuning in an effective manner, therefore is more suited for offline-to-online RL compared to existing conservative learning schemes that directly borrowed from offline RL methods.

**Stability and Optimality**. Performing a trust-region-style update, *PROTO* constrains the policy *w.r.t.* an iteratively evolving policy $\pi_k$, which smartly serves dual purposes by ensuring that: 1) the finetuned policy remains close to the pre-trained policy $\pi_0$ during the initial finetuning stage, to avoid distributional shift; 2) gradually allowing the policy to deviate far from the potentially suboptimal constraint induced by $\pi_0$ at the later stage, to find the optima as long as it stays within the trust region. Therefore, this objective enables stable and optimal policy learning, which is different from and far more optimistic than existing methods with constraints on a potentially suboptimal and fixed $\mu$, $\pi_0$ or $\mathcal{B}$ (Wu et al., 2022; Nair et al., 2020; Kostrikov et al., 2022; Zhang et al., 2023).

Furthermore, we can extend the existing analysis of KL-regularized MDP in the tablular case (corollary of Thm. 1 in (Vieillard et al., 2020)) to our offline-to-online setting and obtain Lemma 1, which shows that *PROTO* principally enjoys both stable and optimal policy finetuning (The derivation is presented in Appendix B). Note that we do not seek to devise tighter and more complex bounds but to give insights for *PROTO*. Following the notations in (Vieillard et al., 2020), we define $Q^*$ is the optimal value of optimal policy $\pi^*$. $\pi_0$ is the pretrained policy and $Q^0$ is its corresponding action-value. Let $v_{\max}^\alpha := \frac{r_{\max}+\alpha\ln|\mathcal{A}|}{1-\gamma}$, $v_{\max} := v_{\max}^0$, and $\epsilon_j$ is the approximation error of value function at $j$-th iteration. Assume that $\|Q^k\|_\infty \leq v_{\max}, k \in N$, then we have:

**Lemma 1.** *Define $Q^k$ as the value of $\pi_k$ obtained at $k$-th iteration by iterating Eq. (4)-(5), then:*

$$\|Q^* - Q^k\|_\infty \leq \frac{2}{1-\gamma}\left\|\frac{1}{k+1}\sum_{j=0}^k \epsilon_j\right\|_\infty + \frac{4}{1-\gamma}\frac{v_{\max}^\alpha}{k+1}, k \in N. \tag{6}$$

For the RHS, the first term reflects how approximation error affects the final performance, and the second term impacts the convergence rate. Note that the approximation error term in Eq. (6) is the norm of average error, *i.e.*, $\|\frac{1}{k+1}\sum_{j=0}^k \epsilon_j\|_\infty$, which might converge to 0 by the law of large numbers. Therefore, *PROTO* will be less influenced by approximation error accumulations and enjoys stable finetuning processes. By contrast, the performance bound of finetuning without any regularization attains the following form (Scherrer et al., 2015) (see Lemma 3 in Appendix for detailed discussion):

$$\|Q^* - Q^k\|_\infty \leq \frac{2\gamma}{1-\gamma}\sum_{j=0}^k \gamma^{k-j}\|\epsilon_j\|_\infty + \frac{2}{1-\gamma}\gamma^{k+1}v_{\max}, k \in N. \tag{7}$$

The error term $\sum_{j=0}^k \gamma^{k-j}\|\epsilon_j\|_\infty \geq 0$ in Eq. (7) cannot converge to 0 and initially decays slowly ($\gamma$ often tends to 1, so $\gamma^k$ changes slowly initially). Therefore, directly finetuning without any regularization may result in severe instability due to the initial approximation error at OOD regions induced during offline pretraining. Previous methods typically introduce additional fixed regularization to stabilize finetuning. However, fixed regularization might lead to a non-eliminable suboptimality gap in the form of (see Lemma 4 in Appendix for detailed discussion):

$$\|Q^* - Q^k\| \leq \frac{\|Q^* - Q_\Pi^*\|_\infty}{1-\gamma}, k \in N, \tag{8}$$

where $Q_\Pi^*$ is the optimal action-value obtained at the constraint set $\Pi$. The RHS of Eq. (8) is hard to converge to 0 unless $\Pi$ contains the optimal policy (Kumar et al., 2019; Wu et al., 2022; Li et al., 2023), but the constraint set $\Pi$ typically only covers suboptimal policies due to the limited coverage of $\mathcal{B}, \mu$ or $\pi_0$. Whereas, the RHS in Eq. (6) can converge to 0 as $k \to \infty$, indicating that the *PROTO* can converge to optimal as $k \to \infty$, which underpins the optimistic nature of *PROTO*.

The comparison between Lemma 1 and Eq. (7)-(8) demonstrates that *PROTO* serves as a seamless bridge between *fixed policy regularization* and *no regularization*, allowing for stability while retaining the optimality of finetuning performance. This indicates that *Iterative Policy Regularization* offers a more reasonable level of conservatism for the offline-to-online RL setting compared to existing policy regularization that directly borrowed from offline RL or no regularization.

**Adaptability and Computational Efficiency.** *PROTO* bridges diverse offline RL/IL and online RL methods, offering a universal proto-framework for offline-to-online RL approaches. It imposes

no assumption on how $\pi_0$ is pretrained and thus can be applied to any offline pretraining method. Also, PROTO can be non-intrusively incorporated into diverse off-policy RL finetuning methods by simply modifying several lines of code to add the regularization term $\log(\pi|\pi_k)$ in the original actor-critic framework, according to Eq. (4)-(5). In addition, calculating the additional regularization term introduces negligible computational cost compared to ensemble networks (Lee et al., 2022) or transformer-based approaches (Zheng et al., 2022), enabling agile and lightweight applications.

### 3.4 PRACTICAL IMPLEMENTATION

To further stabilize the finetuning process and meanwhile retain optimality, we introduce Polyak averaging, a widely adopted technique in modern RL to address potential instability caused by fast target-value update (Haarnoja et al., 2018; Fujimoto et al., 2018), by replacing $\pi_k$ with its delayed updates $\bar{\pi}_k$, *i.e.*, $\bar{\pi}_k \leftarrow \tau\pi_k + (1 - \tau)\bar{\pi}_{k-1}$. Here, $\tau \in (0, 1]$ is a hyper-parameter to control the update speed. As apparent, replacing $\pi_k$ with $\bar{\pi}_k$ retains optimal performance since it still allows large deviation from the pretrained policy (with a slower deviation speed). We also gradually anneal the $\alpha$ value with a linear decay schedule (Wu et al., 2022) for the purpose of weaning off conservatism. Although introducing two hyper-parameters, we show in Appendix D.2 and Appendix F that *PROTO* is robust to changes in hyperparameters within a large range, and parameter tuning can be reduced by adopting a non-parametric approach and setting the annealing speed as a constant.

## 4 EXPERIMENTS

We evaluate on MuJoCo, AntMaze and Adroit tasks with D4RL (Fu et al., 2020) datasets to demonstrate the stable and optimal policy learning, adaptability, and computational efficiency of *PROTO*. Unless otherwise specified, we pretrain policy using a recent SOTA offline RL method EQL (Xu et al., 2023; Garg et al., 2023) for its superior pretraining performances and incorporate the regularization term $\log(\pi|\pi_k)$ from PROTO into SAC (Haarnoja et al., 2018) finetuning for its high sample efficiency and superior performance among off-policy RL methods by default.

### 4.1 BASELINES

We compare *PROTO* with the following baselines: *(i) AWAC* (Nair et al., 2020): an offline-to-online method that implicitly constrains *w.r.t.* the replay buffer $\mathcal{B}$ using AWR-style policy learning. *(ii) IQL* (Kostrikov et al., 2022): a SOTA offline RL method that is also superior in offline-to-online setting since it also utilizes AWR-style policy learning akin to *AWAC*. *(iii) Off2On* (Lee et al., 2022): a SOTA offline-to-online RL method that uses an ensemble of pessimistic value functions together with a balanced experience replay scheme, but it is only applicable for CQL (Kumar et al., 2020) pretraining. *(iv) ODT* (Zheng et al., 2022): a recent decision transformer (Chen et al., 2021) based offline-to-online approach. *(v) PEX* (Zhang et al., 2023): a recent SOTA offline-to-online approach that adaptively constrains the finetuning policy *w.r.t.* the pretrained policy $\pi_0$ by introducing a policy expansion and Boltzmann action selection scheme. *(vi) Offline*: performances of SOTA offline RL approaches without online finetuning that are adopted from (Bai et al., 2022; Xu et al., 2023; Kostrikov et al., 2022; Kumar et al., 2020; Li et al., 2023).

### 4.2 MAIN RESULTS

Learning curves of *PROTO* are illustrated in Figure 2 and 3. Returns are normalized, where 0 and 100 represent random and expert policy performances, respectively. The error bars indicate min and max over 5 different random seeds. Please refer to Appendix D.5 for reproducing details for baselines.

Figure 2 and 3 show that existing policy constraint-based approaches (*IQL*, *AWAC* and *PEX*) in most cases can only marginally outperform or cannot surpass SOTA offline RL approaches, due to the over-conservatism introduced by the policy constraint that largely hinges the finetuning process. This is especially pronounced when offline dataset or pretrained policy is highly-suboptimal such as Adroit manipulation, Antmaze navigation, and MuJoCo locomotion random tasks. In contrast, *PROTO* enjoys both a stable initial finetuning stage and superior final performance owing to the optimistic nature of the proposed iterative policy regularization. Note that *Off2On* also obtains great performance for most MuJoCo locomotion tasks, since it imposes no conservatism during policy finetuning and the tasks are relatively simple. *Off2On*, however, is limited to CQL pretraining, in which case it is hard to yield reasonable performance when the tasks are too difficult for CQL to obtain stable pretrained policies and value functions (*e.g.*, Adroit and Antmaze tasks).

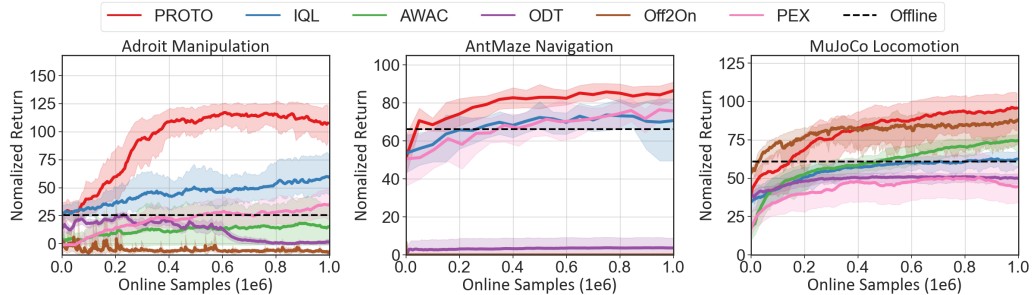

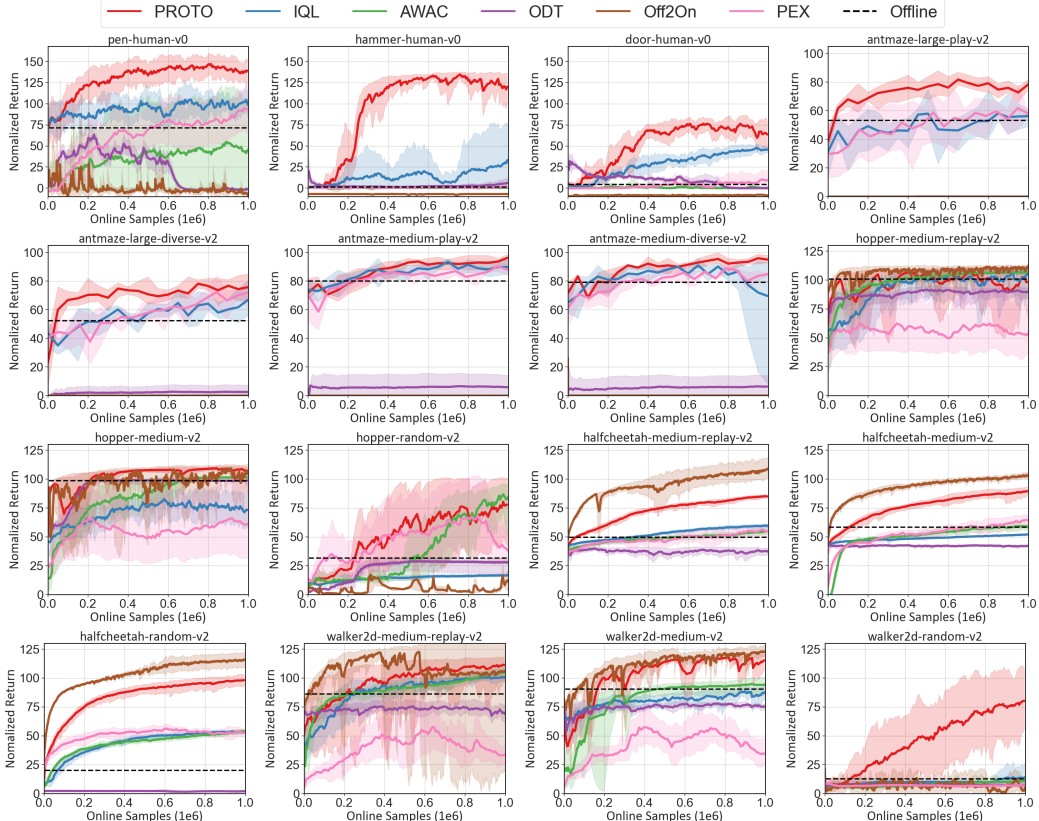

Figure 2: Aggregated learning curves of different approaches on Adroit manipulation, AntMaze navigation, and MuJoCo locomotion tasks from D4RL (Fu et al., 2020) benchmark.

Figure 3: Learning curves of different approaches on Adroit manipulation, AntMaze navigation, and MuJoCo locomotion tasks from D4RL (Fu et al., 2020) benchmark.

### 4.3 EVALUATION ON ADAPTABILITY

To evaluate the universal adaptability of *PROTO*, we train *PROTO* on 4 pretraining and 2 finetuning methods. Such a comprehensive evaluation has not been conducted in previous studies.

**Versatility on Diverse Pretraining Methods**. Except the EQL pretraining, we also pretrain *PROTO* using BC (Pomerleau, 1988), IQL (Kostrikov et al., 2022) and SQL (Xu et al., 2023). *PEX* is the only method that explicitly considers BC pretraining and thus we consider *PEX* with BC pretraining as the main baseline. Figure 4 shows that *BC+PROTO+SAC* surpasses *PEX+BC* by a large margin. Moreover, all *PROTO* variants obtain good results, but solely finetuning without *PROTO* (*EQL+SAC*) suffers from severe performance drop at the initial finetuning stage, demonstrating the crucial role of *PROTO* for offline-to-online. We also observe that *BC+PROTO+SAC* can boost the finetuning performances and obtain good results even starting from an inferior starting point. It is known that offline RL methods are generally hyperparameter-sensitive while BC is much more stable (Kumar et al., 2022). Therefore we can use the simplest BC and bypass the complex offline RL for pretraining.

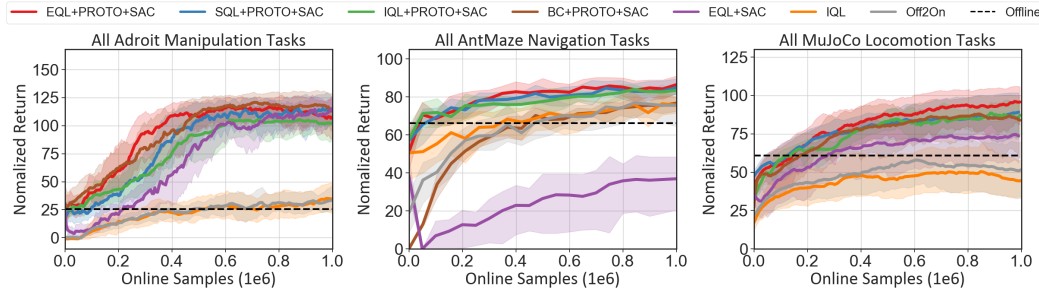

Figure 4: Learning curves of PROTO with different pretraining methods.

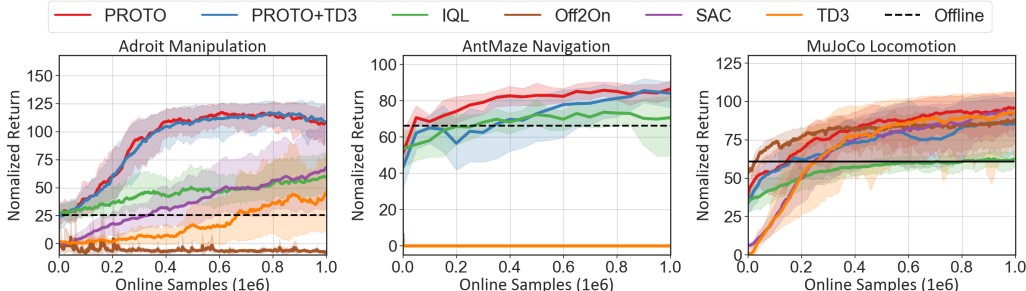

Figure 5: Learning curves of online finetuning for *PROTO+TD3*. See Figure 19 for full results.

**Versatility on Diverse Finetuning Methods**. We also plug *PROTO* into *TD3* (Fujimoto et al., 2018) finetuning. Figure 5 shows that *PROTO+TD3* also obtains SOTA results compared to baselines. Furthermore, we can also extend to sample-efficient online methods via simply increasing the update-to-data (UTD) ratio during finetuning. Due to space limits, please see Appendix E for details.

Altogether, Figure 4 and 5 demonstrate that we can construct competitive offline-to-online RL algorithms by simply combining diverse offline pretraining and online finetuning RL approaches via *PROTO*, which offers a flexible and adaptable framework for future practitioners.

## 4.4 ABLATION STUDY

**Iterative Policy Regularization vs. Fixed Policy Regularization**. To further demonstrate the advantages of *Iterative Policy Regularization*, we replace the iterative policy $\pi_k$ in Eq. (3) with the fixed pretrained policy $\pi_0$ while retains all the other experimental setups and denote this simplest variant as *Frozen*. Similar to previous policy constraint approaches, *Frozen* aims to solve a *fixed policy constrained RL* problem. Figure 6 illustrates the aggregated learning curves of *Frozen* and *PROTO*. We also compare with *IQL* for its strong performances among other baselines in Figure 2.

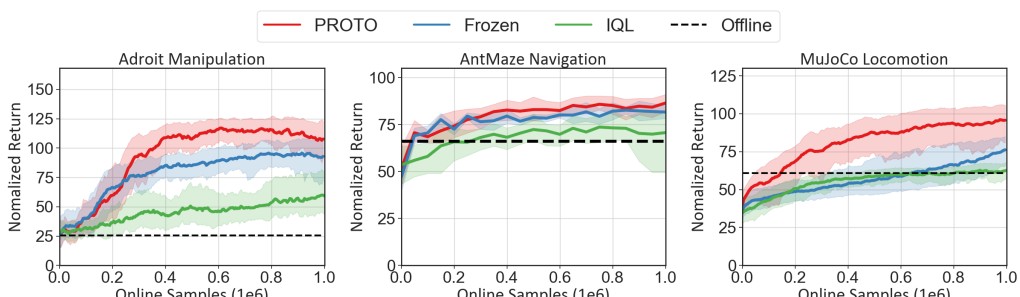

Figure 6: Comparison between iterative policy regularization (*PROTO*) and fixed policy regularization (*Frozen*) and other baselines. Refer to Figure 21 for full results.

Observe in Figure 6 that *PROTO* obtains superior performances compared with *Frozen*, which demonstrates the advantage of *iterative* over *fixed* policy regularization. Note that *Frozen* already annealed the constraint strength, but still converges to suboptimal performances. We believe fixed

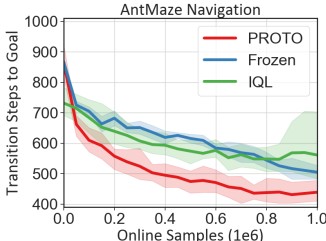
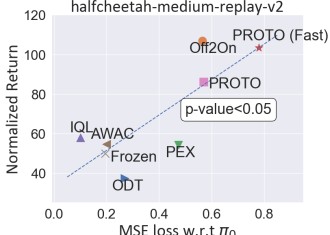
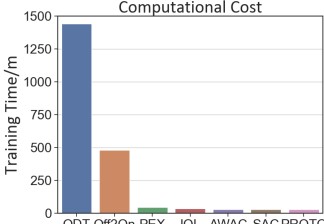

Figure 7: Completion speeds for AntMaze Navigation tasks. Refer to Figure 21 for full results.

Figure 8: Positive correlation between policy deviation *w.r.t.* $\pi_0$ and policy performance.

Figure 9: Computational cost when performing 1M online samples and gradient steps.

regularization requires more relaxed conservatism strength to obtain the optimal results while also being more susceptible to potential approximation errors compared to iterative regularization, please see Appendix C for illustrative explanation. For AntMaze navigation tasks, although *PROTO* obtains similar success rates to other baselines, *PROTO* completes the navigation tasks with much fewer transition steps and higher speed (see Figure 7), translating to much better learned policies.

Also, note that the simplest variant *Frozen* already surpasses or achieves on-par performances as *IQL*. We believe that this is because the in-sample learning paradigm of *IQL* learns from in-sample data only, which lacks supervision on OOD regions and hence hinges exploration. Additionally, we employ a linear decay schedule to wean off conservatism, while the conservatism in *IQL* cannot be fully turned off since *IQL* recovers the maximum of action-value function only when the inverse temperature in its policy extraction step goes to infinity (Kostrikov et al., 2022).

**Finetuning Performance vs. Constraint Strength**. We investigate how constraint strength affects final results of different methods. Figure 8 shows that the final performance and constraint strength exhibit a negative correlation, where *PROTO* attains relaxed constraints and near-optimal performances. Furthermore, we can obtain a better policy by adjusting the polyak averaging speed and conservatism annealing speed, to accelerate the policy deviation speed (reported as *PROTO (Fast)*), which further demonstrates the necessity of relaxing the conservatism when finetuning policies.

We also conduct ablation experiments to analyze the effect of polyak averaging update speed and the conservatism annealing speeds, and find *PROTO* robust to parameter tuning (due to space limits, please refer to Appendix F for detailed results).

## 4.5 COMPUTATIONAL COST

In Figure 9, we report the computation time of performing 1M online samples and gradient steps, to compare the computational efficiency of different methods. It is not surprising that *ODT* requires the most computational resources since it is a transformer-based approach, while other methods build on simple MLPs. *Off2On* requires an ensemble of pessimistic Q-functions and a complex balanced experience replay scheme, which imposes high computational cost. In addition, the CQL pretraining in *Off2On* explicitly requires performing computationally-expensive numerical integration to approximate the intractable normalization term in continuous action spaces (Kumar et al., 2020; Kostrikov et al., 2021). By contrast, *PROTO* only requires calculating the additional regularization term, computational overhead of which is negligible, therefore enjoys the same computational efficiency as standard off-policy RL methods.

## 5 CONCLUSION AND FUTURE WORK

To address major drawbacks of existing offline-to-online RL methods (suboptimal performance, limited adaptability, low computational efficiency), we propose *PROTO* that incorporates an iteratively evolved regularization term to stabilize the initial finetuning and bring enough flexibility to yield strong policies. *PROTO* seamlessly bridges diverse offline RL/IL and online off-policy RL methods with a non-intrusively modification, offering a flexible and efficient offline-to-online RL proto-framework. Following existing works, this paper only focuses on off-policy RL finetuning, which has high-sample efficiency but may not enjoy monotonic policy improvement guarantees. One appealing future direction is to introduce *PROTO* into on-policy RL finetuning or marry off-policy sample efficiency with on-policy monotonic improvement to devise more advanced offline-to-online RL.

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

# A  DETAILED DISCUSSIONS ON RELATED WORKS

This section provides detailed comparisons with existing practical offline-to-online RL methods in Table 2.

Table 2: Detailed comparisons with related practical offline-to-online RL methods. $\mu$: behavior policy that generates the offline dataset $\mathcal{D}$. $\mathcal{B}$: replay buffer. $\pi_0$: pretrained policy. a. Constraint policy set; b. Stable and optimal policy learning; c. Adaptable to diverse pretraining methods; d. Adaptable to diverse finetuning methods; e. Computational efficient.  ✓: Yes, ✗: No, ○: It depends.

| Type | Method | a. | b. | c. | d. | e. |
|---|---|---|---|---|---|---|
| PC | SPOT (Wu et al., 2022) | $\mu$ | ✗ | ✓ | ✓ | ✓ |
|  | AWAC (Nair et al., 2020) | $\mathcal{B}$ | ✗ | ✓ | ✗ | ✓ |
|  | IQL (Kostrikov et al., 2022) |  |  |  |  |  |
|  | XQL (Garg et al., 2023) |  |  |  |  |  |
|  | InAC (Xiao et al., 2023) |  |  |  |  |  |
|  | O3F (Mark et al., 2023) |  |  |  |  |  |
|  | PEX (Zhang et al., 2023) | $\pi_0$ | ✗ | ✓ | ✓ | ✓ |
|  | ACA (Yu & Zhang, 2023) | $\pi_0$ | ✗ | ✗ | ✗ | ✓ |
| PVI | Off2On (Lee et al., 2022) | No Constraint | ✓ | ✗ | ✓ | ✗ |
|  | Cal-QL (Nakamoto et al., 2023) |  |  |  |  |  |
|  | MCQ (Lyu et al., 2022) |  |  |  |  |  |
|  | CCVL (Hong et al., 2023) |  |  |  |  |  |
| GCSL | ODT (Zheng et al., 2022) | $\mathcal{B}$ | ✗ | ✗ | ✗ | ✗ |
| Others | APL (Zheng et al., 2023) | ○ | ○ | ○ | ✗ | ○ |
|  | SUNG (Guo et al., 2023) |  |  |  |  |  |
|  | E2O (Zhao et al., 2023) |  |  |  |  |  |
|  | PROTO (Ours) | $\pi_k$ | ✓ | ✓ | ✓ | ✓ |

**Stable and optimal policy learning**. All existing policy constraint and goal conditioned supervised learning methods constrain the finetuning policy *w.r.t.* a fixed policy set induced by the behavior policy $\mu$, replay buffer $\mathcal{B}$ or the pretrained policy $\pi_0$, which may be highly-suboptimal and induce large optimality gap. APL (Zheng et al., 2023) and SUNG (Guo et al., 2023) finetune the policy with the existence of pretraining conservatism, which may also lead to suboptimal performances. E2O (Zhao et al., 2023) is a concurrent work that finetunes an ensemble of pretrained agents to stabilize offline-to-online performances, which can apply to diverse pretraining methods. Therefore, E2O can obtain optimal performances when the conservatism term can be fully dropped such as the value regularization in CQL (Kumar et al., 2020) and the BC term in TD3+BC (Fujimoto & Gu, 2021). Otherwise, E2O cannot obtain optimal results, such as using IQL pretraining since the conservatism of IQL cannot be fully dropped during online finetuning. By contrast, PROTO casts constraint on an iteratively evolving constraint set induced by the policy $\pi_k$ at the last iteration, which relaxes the conservatism in the later stage and thus enjoys similar optimality as no constraints.

**Adaptability to diverse finetuning and pretraining methods**. SPOT (Wu et al., 2022) provides a pluggable policy regularization as PROTO dose, thus can flexibly extend to diverse methods, but requires the hard estimation of the unknown behavior policy. AWAC (Nair et al., 2020), IQL (Kostrikov et al., 2022), XQL (Garg et al., 2023), InAC (Xiao et al., 2023) use AWR to extract policy without the need for explicitly behavior policy estimation. AWR, however, is difficult to plug in other online RL approaches non-intrusively, limiting its adaptability for diverse online finetuning methods. PEX

offers the most flexible framework among all the baselines and is adaptable to diverse methods. Off2On (Lee et al., 2022), Cal-QL (Nakamoto et al., 2023), MCQ (Lyu et al., 2022), CCVL (Hong et al., 2023) can only apply to CQL-style (Kumar et al., 2020) pretraining, but can finetune with diverse online RL approaches. ODT (Zheng et al., 2022) is specifically designed for DT (Chen et al., 2021) finetuning and pretraining, and meanwhile ACA (Yu & Zhang, 2023) is specifically designed for SAC+BC pretraining and SAC finetuning. Therefore, ODT and ACA suffer from the most limited applicability. APL and SUNG also offer general frameworks for offline-to-online RL, but can only apply to offline RL methods that have explicit regularization terms such as TD3+BC and CQL, since they require to adaptively drop the regularization terms during online finetuning (e.g. the BC term in TD3+BC and the value regularization term in CQL). The online finetuning methods of E2O is also restricted to the choice of its offline pretraining methods, suffering from limited adaptability.

**Computational efficiency**. Off2On and E2O requires an ensemble of value functions and hence is computation inefficient. Moreover, Off2On pretrains using CQL, which explicitly requires performing computationally-expensive numerical integration to approximate the intractable normalization term in continuous action spaces (Kumar et al., 2020; Kostrikov et al., 2021), which inevitably introducing tremendous computational costs. ODT builds on Transformer (Vaswani et al., 2017) architecture, which requires far more computational resources compared to other methods that builds on simple MLPs.

**Other relevant works**. TD3-C (Luo et al., 2023) is a relevant concurrent work that only focuses on TD3+BC pretraining and TD3 finetuning, which can be perceived as one variant of our PROTO framework (*PROTO+TD3*) and does not provide theoretical interpretation to their methods. Trust-PCL (Nachum et al., 2018) is a relevant work that also solves a KL-regularized MDP, but focuses only on online RL setting without the offline pretraining. QDagger (Agarwal et al., 2022) is another recent work that constrains on the fixed pretrained policy $\pi_0$, but is designed for DQN (Mnih et al., 2015)-based methods and focuses on discrete control. BREMEN (Matsushima et al., 2020) is another work that utilizes trust-region updates but focuses on the deployment-efficient setting and also casts constraints w.r.t the replay buffer $\mathcal{B}$, which is over-conservative. There are also some works that utilize trust-region updates to exclusively address the online RL (Janner et al., 2019; Schulman et al., 2017) or the offline RL problems (Siegel et al., 2019; Zhuang et al., 2023).

There is also a large amount of work focusing on accelerating online RL training using offline dataset (Ball et al., 2023), representation learning (Laskin et al., 2020), or a guiding policy (Uchendu et al., 2023). A recent work (Niu et al., 2022) also considers the hybrid offline and online RL setting but aims to tackle the sim2real gap. There are also some theoretical works that focus on the hybrid offline and online RL setting (Wagenmaker & Pacchiano, 2023; Song et al., 2023) or the offline-to-online RL setting (Xie et al., 2021) without practical implementation. These works lie in the orthogonal scope of our paper and thus we do not provide detailed discussions here.

Based on the above thorough discussions, PROTO is the only work that can simultaneously enjoy all the good properties including stable and optimal policy learning, adaptability to diverse methods and high computational efficiency. In addition, similar to all these related works, this paper contains no potential negative societal impact.

# B THEORETICAL INTERPRETATIONS

This section shows how to extend the analysis of KL-regularized MDP (Vieillard et al., 2020) to our offline-to-online RL setting (Lemma 1) and provides detailed discussions on the inherent stability and optimality of *PROTO*. Specifically, we consider the tabular case to ease the analysis, which is a common treatment in previous works (Ma et al., 2022; Lyu et al., 2022; Xiao et al., 2023; Vieillard et al., 2020). In detail, we can extend existing theories from the online RL setting to our specific offline-to-online settings to compare between different types of policy regularizations, including iterative/fixed/no policy regularizations in terms of finetuning stability and optimality. Note that we do not seek to devise tighter and more complex bounds but to give insightful interpretations for our proposed approach *PROTO*.

## B.1 PROOF OF LEMMA 1

*Proof.* First, we introduce Lemma 2 (Corollary of Theorem 1 in (Vieillard et al., 2020)), which builds the foundation of our theoretical interpretation.

**Lemma 2.** *Define $Q^*$ is the action-value of optimal policy $\pi^*$, $Q^k$ is the action-value of policy $\pi_k$ obtained at $k$-th iteration by iterating Eq. (4)-(5). $v_{\max}^\alpha := \frac{r_{\max} + \alpha \ln|\mathcal{A}|}{1-\gamma}$. $\epsilon_j$ is the approximation error of the action-value function. Assume that $\pi_0$ is a uniform policy, $Q^0$ is initialized such that $\|Q^0\|_\infty \leq v_{\max}$ and $\|Q^k\|_\infty \leq v_{\max}$. We have (Vieillard et al., 2020):*

$$\|Q^* - Q^k\|_\infty \leq \frac{2}{1-\gamma} \left\| \frac{1}{k} \sum_{j=1}^{k} \epsilon_j \right\|_\infty + \frac{4}{1-\gamma} \frac{v_{\max}^\alpha}{k}, k \in N^+. \tag{9}$$

At first glance, Lemma 2 is quite similar to Lemma 1. However, note that the assumption of Lemma 2 is slightly different from Lemma 1. With a slight abuse in notation, in Lemma 2, $\pi_0$ is assumed as a uniform policy and $Q^0$ can be any initialized action-value function that satisfies $\|Q^0\|_\infty \leq v_{\max}$, while in Lemma 1, $\pi_0$ is the pretrained policy and $Q^0$ is the corresponding action-value. This means that Lemma 2 is conducted on the pure online RL setting. Nevertheless, we will show that this lemma can seamlessly extend to Lemma 1 and our offline-to-online setting by introducing an additional negligible requirement on $Q^0$ initialization.

Since the original assumption in Lemma 2 allows any form of $Q^0$ initialization as long as it satisfies $\|Q^0\|_\infty \leq v_{\max}$, we introduce one additional constraint on $Q^0$ initialization that the pretrained policy should be obtained via one policy improvement step upon $Q^0$, *i.e.*, $\arg\max_\pi \mathbb{E}_{a\sim\pi(\cdot|s)}[Q^0(s,a) - \alpha \cdot \log\left(\frac{\pi(s,a)}{\pi_0(s,a)}\right)$. This assumption is negligible since it only introduce an additional condition under the premise of satisfying the original assumption.

Under this mild assumption, the pretrained policy and its action-value in offline-to-online setting become $\pi_1$ and $Q^1$ in Lemma 2, respectively. Note that the pretrained policy is $\pi_0$ and its corresponding action-value function is $Q^0$ in Lemma 1. Therefore, the conclusion of Lemma 2 is ready to transfer to Lemma 1 with some simple modifications to meet the requirement that $k = 1$ in Lemma 2 equivalents to $k = 0$ in Lemma 1:

**Lemma 1**. *Define $Q^k$ as the action-value of policy $\pi_k$ obtained at $k$-th iteration by iterating Eq. (4)-(5), and $Q^*$ is the optimal value of optimal policy $\pi^*$. $\pi_0$ is the pretrained policy and $Q^0$ is its corresponding action-value function. Let $v_{\max}^\alpha := \frac{r_{\max} + \alpha \ln|\mathcal{A}|}{1-\gamma}$, and $\epsilon_j$ is the approximation error of the action-value function. Assume that $\|Q^k\|_\infty \leq v_{\max}^0$, we have:*

$$\|Q^* - Q^k\|_\infty \leq \frac{2}{1-\gamma} \left\| \frac{1}{k+1} \sum_{j=0}^{k} \epsilon_j \right\|_\infty + \frac{4}{1-\gamma} \frac{v_{\max}^\alpha}{k+1}, k \in N. \tag{10}$$

$\square$

## B.2 STABILITY AND OPTIMALITY COMPARED WITH PREVIOUS METHODS

**Stability.** The approximation error term in Eq. (10) is the infinity norm of the average estimation errors, *i.e.*, $\|\frac{1}{k+1}\sum_{j=0}^{k} \epsilon_j\|_\infty$, which can converge to 0 by the law of large numbers. This indicates that *PROTO* will be less influenced by approximation error accumulations and enjoys stable finetuning processes, owing to the stabilization of iterative policy regularization.

Then, we recall the typical approximation error propagation without any regularization in Lemma 3 (can be found at Section 4 in (Vieillard et al., 2020) and (Scherrer et al., 2015)).

**Lemma 3.** *Assume $\pi_0$ is a uniform policy, $Q^0$ is initialized such that $\|Q^0\|_\infty \leq v_{\max}$ and $\|Q^k\|_\infty \leq v_{\max}$, where $v_{\max} := \frac{r_{\max}}{1-\gamma}$. Then we have*

$$\|Q^* - Q^k\|_\infty \leq \frac{2\gamma}{(1-\gamma)^2} \left( (1-\gamma) \sum_{j=1}^{k} \gamma^{k-j} \|\epsilon_j\|_\infty \right) + \frac{2}{1-\gamma} \gamma^k v_{\max}, k \in N^+. \tag{11}$$

Similar to the difference between Lemma 2 and Lemma 1, Lemma 3 cannot be directly applied in the offline-to-online RL setting (Eq. (7)). However, it can be easily transferred by imposing a minimal assumption on $Q^0$ initialization akin to the proof of Lemma 1. We will not elaborate on this again and instead directly focus on its approximation error term. As shown in Lemma 3 or Eq. (7), the approximation error term is the discounted sum of the infinity norm of the approximation error at each iteration $\sum_{j=1}^{k} \gamma^{k-j} \|\epsilon_j\|_\infty$, which cannot converge to 0 and thus is non-eliminable. Furthermore, this term often initially decays slowly since $\gamma$ usually tends to 1. Therefore, if the initial approximation error $\epsilon_0$ caused by offline pretraining is pretty large, the effects of $\gamma^k \|\epsilon_0\|_\infty$ might cause severe instability to the initial finetuning (when k is small). This explains why directly finetuning an offline pretrained policy with online RL typically leads to an initial performance drop and requires additional regularization to stabilize the training process.

**Optimality**. To stabilize the online finetuning, previous policy constraint based offline-to-online RL approaches typically constrain the finetuning policy in a fixed constraint set $\Pi$ that is induced by the behavior policy $\mu$, the pretrained policy $\pi_0$ or the replay buffer $\mathcal{B}$[1]. However, as discussed in previous works (Kumar et al., 2019; Li et al., 2023; Wu et al., 2022), optimizing in a fixed constraint set typically lead to large optimality gap as Lemma 4 (Theorem 4.1 in (Kumar et al., 2019), Theorem 3 in (Li et al., 2023)) shows:

**Lemma 4.** *(Suboptimality induced by fixed policy regularization). Define $Q_\Pi^*$ is the optimal value obtained at a constrained policy set $\Pi$. $Q^k$ is the action-value of policy $\pi_k$ obtained at k-th iteration by iterating Eq. (1)-(2), but in the set $\Pi$, i.e., $\pi_k \leftarrow \arg\max_{\pi \in \Pi} \mathbb{E}_{a \sim \pi}[Q^{k-1}(s, a)]$. Then we always have a suboptimality gap:*

$$\|Q^* - Q^k\|_\infty \leq \frac{\|Q^* - Q_\Pi^*\|_\infty}{1 - \gamma}, k \in N \tag{12}$$

Note that the RHS of Eq. (12) has only an implicit correlation with the iteration times $k$ since the policy set $\Pi$ will gradually expand when the replay buffer $\mathcal{B}$ contains more data or when the conservatism strength is fully decayed. However, $\Pi$ varies slowly since $\mathcal{B}$ is typically large and will be less effected by filling in a few of data and solely annealing the conservatism strength requires more relaxation as shown in Figure 10. Therefore, the suboptimality gap is hard to be eliminated, which may cause a large suboptimality gap when the constraint policy set $\Pi$ is highly suboptimal.

On the contrary, Lemma 1 provides an intuitive insight that the suboptimality gap can be fastly minimized to zero as $k$ grows even with the existence of iterative policy regularization. Therefore, *PROTO* can be perceived as an "optimistic" conservatism, which will not lead to suboptimal performances caused by over-conservatism. This can be observed in Figure 6 that constraining on a fixed potentially suboptimal constraint set may result in suboptimal performances while allowing the constraint set to actively update can yield near-optimal performances. It is also worth mentioning that the complete version of Eq. (12) contains how approximation error and distributional shifts affect final performances, but we leave them behind to ease the readers to catch the main difference between fixed policy regularization and iterative policy regularization.

## C    INTUITIVE ILLUSTRATION OF ITERATIVE POLICY REGULARIZATION

To alleviate the over-conservatism caused by fixed policy regularization, previous studies typically anneal the conservatism strength. However, observe from Figure 6, Figure 20 and Figure 21 that even with the conservatism annealing, fixed policy regularization still underperforms iterative policy regularization. Apart from the theoretical interpretation, we give an intuitive illustration in Figure 10 to further explain why simply annealing the conservatism strength is not enough to obtain optimal policy.

---

[1]The constraint set induced by $\mathcal{B}$ slowly changes with filling in new transitions, but is much slower than the one induced by $\pi_k$.

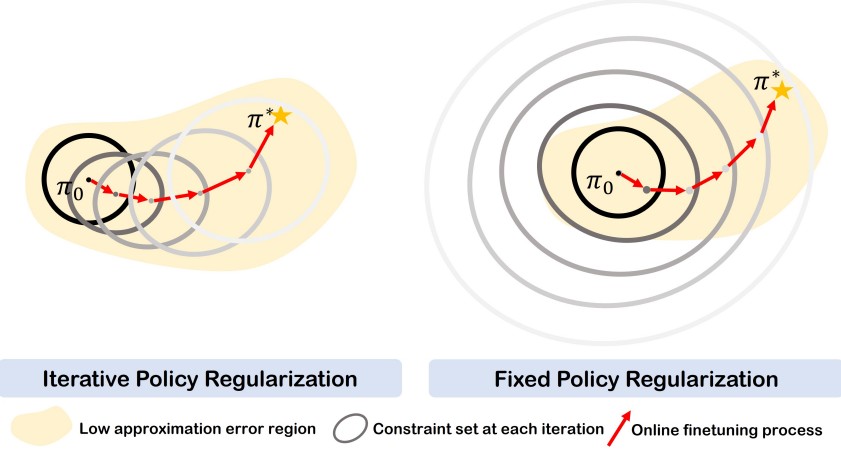

Figure 10: Illustration of iterative policy regularization *v.s* fixed policy regularization. In order to achieve the same optimal policy, fixed policy regularization requires a relaxation of conservatism strength while also being more susceptible to potential approximation errors compared to iterative policy regularization.

In Figure 10, the size of the constraint set represents the conservatism strength determined by $\alpha$. A larger $\alpha$ induces stronger conservatism, resulting in a strict and small constraint set, while a smaller $\alpha$ implies a more relaxed conservatism and a larger constraint set.

As depicted in Figure 10, to achieve optimal performance, iterative policy regularization gradually deviates from the initial policy while maintaining relatively less-changed conservatism strength. In contrast, fixed policy regularization requires significantly relaxed conservatism to attain the same optimal performance level. Moreover, this relaxed conservatism includes more regions that experience large approximation errors, rendering online finetuning more susceptible to potential approximation errors compared to iterative policy regularization.

## D    EXPERIMENTAL DETAILS

This section outlines the experimental details to reproduce the main results in our paper. We'll also open source our code for other researchers to better understand our paper.

### D.1    TASK DESCRIPTION

**Adroit Manipulation**. Adroit manipulation contains 3 domains: *pen, door, hammer*, where the RL agent is required to solve dexterous manipulation tasks including rotating a pen in specific directions, opening a door, and hammering a nail, respectively. The offline datasets are *human-v0* datasets in D4RL (Fu et al., 2020) benchmark, which only contain a few successful non-markovian human demonstrations and thus is pretty difficult for most offline RL approaches to acquire reasonable pretraining performances.

**Antmaze Navigation**. Antmaze navigation consists of two domains, namely *medium* and *large*, each with two datasets from the D4RL (Fu et al., 2020) benchmark: *play-v2* and *diverse-v2*. In each domain, the objective is for an ant to learn how to walk and navigate from the starting point to the destination in a maze environment, with only sparse rewards provided. This task poses a challenge for online RL algorithms to explore high-quality data effectively without the support of offline datasets or additional domain knowledge.

**MuJoCo Locomotion**. MuJoCo locomotion encompasses several standard locomotion tasks commonly utilized in RL research, such as *Hopper, Halfcheetah, Walker2d*. In each task, the RL agent is tasked with controlling a robot to achieve forward movement. The D4RL (Fu et al., 2020) benchmark provides three types of datasets with varying quality for each task: *random-v2, medium-v2, medium-replay-v2*.

## D.2 HYPER-PARAMETERS

**Online finetuning hyper-parameters.** *PROTO* has two hyper-parameters: the Polyak averaging speed $\tau$ and the conservatism linear decay speed $\eta$. Although with 2 hyper-parameters, we find that choosing $\eta$ around 0.9 can achieve substantially stable performances for all 16 tasks (see Figure 15 for detailed results), where $\eta = 1$ means $\alpha$ anneals to 0 with $10^6$ online samples and $\eta = 0.9$ means $\alpha$ anneals to 0 with $\frac{10^6}{0.9}$ online samples. Therefore, we adopt a non-parametric treatment by setting $\eta = 0.9$. We report the detailed setup in Table 3.

Table 3: Online finetuning hyper-parameters

| Task | $\tau$ | $\eta$ |
|------|--------|--------|
| All MuJoCo locomotion | $5e-3$ | 0.9 |
| All Antmaze navigation | $5e-5$ | 0.9 |
| All Adroit manipulation | $5e-5$ | 0.9 |

Table 4: Offline pretraining hyper-parameters

| Task | Initial $\alpha$ | Pretraining Steps |
|------|------------------|-------------------|
| All MuJoCo locomotion | 2.0 | 0.1M |
| All Antmaze navigation | 0.5 | 0.2M |
| All Adroit manipulation | 2.0 | 0.1M |

Table 3 shows that *PROTO* has only two groups of hyper-parameters across 16 tasks with only varying the Polyak averaging speed $\tau$. Moreover, we ablate on the Polyak averaging speed $\tau$ in the range of $[0.5\tau, 2\tau]$ in Figure 13 and find that *PROTO* is robust to such large hyper-parameter variations. As a result, *PROTO* is easy to tune, which is critical and desired for RL community since online evaluations for parameter tuning are generally costly.

**Offline pretraining hyper-parameters.** In our paper, we pretrain the policy using EQL (Xu et al., 2023) (equivalents to XQL (Garg et al., 2023)), and hence we directly adopt the conservatism strength coefficient $\alpha$ in EQL paper (Xu et al., 2023) to pretrain policy and use it to initialize $\alpha$ in our paper. In terms of pretraining steps, We find that performing 0.1M pretraining steps for all MuJoCo locomotion and Adroit manipulation tasks and 0.2M pretraining steps for all Antmaze navigation tasks already attain good initialization. Therefore, we do not pretrain further to reduce computational costs. Please see Table 4 for detailed parameter choice[2]. Similar to EQL, both PEX and IQL pretrain based on in-sample learning. Therefore, we also pretrain PEX and IQL with the pretraining steps according to Table 4.

## D.3 ADDITIONAL EXPERIMENTAL DETAILS

**Initialization of online replay buffer.** We initialize the online replay buffer with three different types: (1) Initialize the buffer with the entire offline dataset akin to (Nair et al., 2020; Kostrikov et al., 2022). (2) Conduct a separate online buffer and sample symmetrically from both offline dataset and online buffer akin to (Ball et al., 2023). (3) Initialize the buffer with a small set of offline datasets. However, we observe that these three different types of initialization have little effect on finetuning performances, please see Table 5 for detailed results. Whereas, we recommend to symmetrically sample from offline dataset and together a separate online dataset akin to (Ball et al., 2023) for future consideration to design online finetuning approaches with higher sample efficiency.

We believe that PROTO is less affected by the replay buffer compared to other methods since PROTO constrains the policy based on the iteratively evolved policy $\pi_k$ rather than relying heavily on the

---

[2]We also initialize $\alpha$ according to Table 4 when using BC to pretrain policy.

Table 5: Training results with different kinds of online replay buffer

| | Initialize online buffer using full offline data | Symmetric sample akin to RLPD (Ball et al., 2023) | Initialize online buffer using small offline data |
|---|---|---|---|
| Mujoco mean | 95.5 | 96.7 | 94.2 |
| Antmaze mean | 86.0 | 83.4 | 86.4 |
| Adroit mean | 113.2 | 115.1 | 117.6 |

replay buffer. On the other hand, the conservatism in other methods like APL (Zheng et al., 2023) and Off2On (Lee et al., 2022) is largely influenced by the replay buffer (e.g., the BC term in TD3+BC and the value regularization term in CQL depend heavily on the replay buffer). Therefore, PROTO exhibits a replay-buffer-agnostic behavior, which is advantageous as it reduces the burden of designing ad-hoc replay buffer update strategies to achieve good results.

**Network architecture and optimization hyper-parameters**. We implement all the function approximators with 2-layer MLPs with ReLU activation functions. To stabilize both offline pretraining and online finetuning processes, we add Layer-Normalization (Ba et al., 2016) to the action-value networks and state-value networks akin to previous works (Xu et al., 2023; Garg et al., 2023; Ball et al., 2023). We find that Layer-Normalization may cause over-conservatism for all halfcheetah tasks, and thus we only drop Layer-Normalization when experimenting on all halfcheetah tasks. We choose Adam (Kingma & Ba, 2015) as optimizer, 3e-4 as learning rate and 256 as batch size for all networks and all tasks.

**Clip-double Q and value regularization backup**. Similar to (Ball et al., 2023), we also find that clip-double Q and the value regularization backup may introduce over-conservatism and cause inferior performances for some extremely difficult tasks. Therefore, we do not use these trick for some experiments as Table 6 shows.

Table 6: Additional experiment details

| Task | Clip-double Q | Value regularization backup |
|---|---|---|
| All Mujoco-Locomotion | ✓ | ✓ |
| All Antmaze-Navigation | ✗ | ✗ |
| All Adroit-Manipulation | ✓ | ✗ |

### D.4 Pseudocode and computational cost

This subsection presents the pseudocode when finetuning using SAC.

---

**Algorithm 1** PROTO with SAC finetuning

---

**Input:** Offline dataset $\mathcal{D}$, online replay buffer $\mathcal{B}$, pretrained value networks $Q^0$, pretrained policy $\pi_0$, initial conservatism strength $\alpha$.
  **for** $k = 0, 1, 2, 3, ..., N$ **do**
    Collect new transition, $\mathcal{B} \leftarrow \mathcal{B} \cup \{(s, a, r, s')\}$.
    Sample mini-batch transitions $B \sim \mathcal{D} \cup \mathcal{B}$.
    Update SAC action-value networks based on $B$ by subtracting $\alpha \cdot \log \frac{\pi}{\pi_k}$ from target value.
    Update SAC policy based on $B$ by adding $\alpha \cdot \log \frac{\pi}{\pi_k}$ from from actor loss.
    Update target value networks via polyak averaging trick
    Update target actor network $\bar{\pi}_k$ via polyak averaging trick
    Anneal $\alpha$ until 0.
  **end for**

---

We implement our approach using the JAX framework (Bradbury et al., 2018). On a single RTX 3080Ti GPU, we can perform 1 million online samples and gradient steps in approximately 20 minutes for all tasks.

### D.5 BASELINE REPRODUCING DETAILS

We rerun the official codes and adhere to the authors' reported hyperparameters to reproduce most of the baseline results. AWAC results are reproduced by the open-sourced d3rlpy repo (Seno & Imai, 2022). Note that most of the baselines have far more hyperparameter tuning than PROTO.

1. **IQL**: https://github.com/ikostrikov/implicit_q_learning. We use 3 groups of hyperparameters via tuning the expectile value, policy temperature and dropout rate following the official implementations. Specifically, the expectile value is 0.7 for Mujoco tasks, 0.9 for Antmaze tasks, 0.7 and for Adroit tasks. The policy temperature is 3.0 for Mujoco tasks, 10.0 for Antmaze tasks, and 0.5 for Adroit tasks. The dropout rate for Adroit tasks is 0.1 and 0 for other tasks since the offline data for Adroit is small and is prone to overfitting.

2. **ODT**: https://github.com/facebookresearch/online-dt. We tune various hyperparameters following the official implementations including learning rate, weight decay, pretrain step and whether to use position encoding, et, al, leading to more than 10 groups of parameters. Please refer to the Table C.4 in (Zheng et al., 2022) for details. In addition, ODT does not conduct experiments on Antmaze-medium and Antmaze-large tasks, thus we adopt the hyperparameters used for Antmaze-umaze to task to run Antmaze-medium and large tasks. Also, ODT does not evaluate on Adroit tasks. For Adroit tasks, we set the pretraining steps as 5000, the buffer size as 1000, the learning rate as 1e-4, the weight decay as 5e-4, the eval context length as 5, $g_{\text{eval}}$ as the expert score of each task reported in (Fu et al., 2020) and $g_{\text{online}}$ as the double of the expert score of each task reported in (Fu et al., 2020), the position encoding is set to NONE.

3. **PEX**: https://github.com/Haichao-Zhang/PEX. We use the same 3 groups of hyperparameters as IQL since the hyperparameters reported in the PEX paper are directly adopted from IQL.

4. **Off2On**: https://github.com/shlee94/Off2OnRL. Off2On requires tuning 2 groups of hyperparameters for pretraining on Mujoco, Antmaze, and Adroit tasks, focusing on the Lagrangian threshold following the official implementations. Specifically, the Lagrangian threshold for Mujoco and Adroit tasks is -1.0 and is 5.0 for Antmaze tasks. We pretrain using the official CQL implementation[3].

5. **AWAC**: https://github.com/takuseno/d3rlpy We reproduce the AWAC using the open-sourced library from d3rlpy (Seno & Imai, 2022).

## E SAMPLE-EFFICIENT ONLINE FINETUNING

In this section, we show that PROTO can be also extended to sample-efficient online finetuning setting by simply increasing the update-to-data (UTD) ratio from 1 to 20 and obtain comparable results compared to the recent SOTA sample-efficient baselines (RLPD[4] (Ball et al., 2023)). In the sample-efficient online finetuning setting, we only finetune with 0.1M rather than 1M online samples. The aggregated learning curves is presented in Figure 11.

Figure 11 shows that PROTO achieves on-par performances compared to RLPD via simply increasing the UTD ratio (PROTO (UTD=20)), further demonstrating the adaptability of PROTO on sample-efficient online finetuning. However, we observe that RLPD grows faster than PROTO. We hypothesize that this is because that PROTO requires conservatism to handle the instability at the initial finetuning stage, which may slow down the initial policy updates. In contrast, RLPD only focuses on online RL without offline pretraining, and thus can ignore the conservatism and enjoy fast updates. The theoretical comparison between Eq. (7) and Eq. (6) also shows that the KL-regularization in PROTO will change the optimality gap convergence rate from $\gamma^k$ (Eq. (7)) to $\frac{1}{k+1}$ (Eq. (6)), which means the convergence becomes slower. Moreover, PROTO uses only a single network but RLPD utilizes an ensemble network that ensembles 10 networks. Ensemble network is

---

[3] https://github.com/aviralkumar2907/CQL
[4] https://github.com/ikostrikov/rlpd

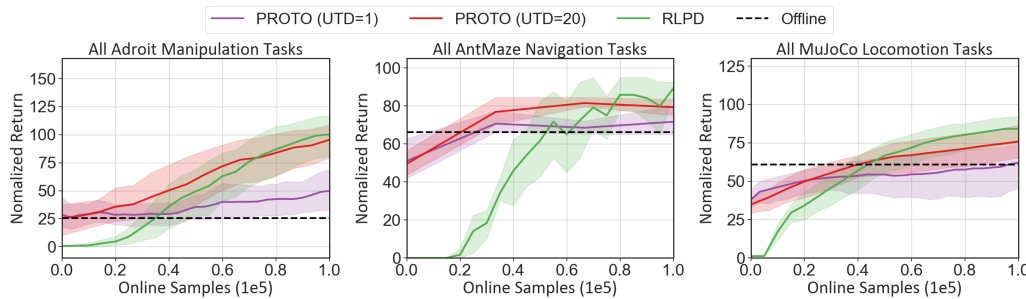

Figure 11: Comparisons with the SOTA sample-efficient RL method, RLPD (Ball et al., 2023).

believed to perform better than a single network but requires more computational resources (Ball et al., 2023; Lee et al., 2022).

### E.1 COMPARISONS WITH NON-OPEN-SOURCED SAMPLE-EFFICIENT BASELINES

We also compare PROTO with other non-open sourced baselines including APL (Zheng et al., 2023) and SUNG (Guo et al., 2023). We also finetune with only 0.1M online samples and report the results in Table 7, the results of APL and SUNG are adopted from (Guo et al., 2023).

Table 7: Comparison with the non-open-sourced sample-efficient baselines, including all varaints of APL (Zheng et al., 2023) and SUNG (Guo et al., 2023). The top 3 scores for each task are bolded.

| | APL (TD3+BC) | APL (CQL) | APL (SPOT) | SUNG (TD3+BC) | SUNG (CQL) | SUNG (SPOT) | PROTO (UTD=20) | PROTO (UTD=20, $10\tau$) |
|---|---|---|---|---|---|---|---|---|
| antmaze-m-p | - | 22.8 | 86.0 | - | 86.3 | **88.6** | **92.3** | **88.3** |
| antmaze-m-d | - | 36.8 | 86.0 | - | 85.6 | **91.7** | **90.5** | **89.3** |
| antmaze-l-p | - | 0.0 | 38.9 | - | **52.7** | 45.7 | **70.9** | **78.7** |
| antmaze-l-d | - | 0.0 | 3.8 | - | **44.1** | 19.8 | **70.7** | **70.0** |
| Antmaze Total | - | 59.6 | 214.7 | - | 268.7 | 245.8 | **324.4** | **326.3** |
| hopper-r | 27.1 | 41.8 | - | 38.7 | **44.3** | - | 43.9 | **69.9** |
| halfcheetah-r | 70.0 | 67.7 | - | **76.6** | 69.1 | - | **84.9** | **83.7** |
| walker2d-r | 13.8 | 6.3 | - | 14.1 | **14.5** | - | **49.4** | **39.9** |
| hopper-m | 76.9 | **102.7** | - | 101.8 | **104.1** | - | **105.3** | 99.9 |
| halfcheetah-m | **80.9** | 44.7 | - | **80.7** | **79.7** | - | 73.4 | 78.5 |
| walker2d-m | 98.2 | 75.3 | - | **113.5** | 86.0 | - | **107.0** | **113.1** |
| hopper-m-r | **100.6** | 97.4 | - | **101.3** | **101.9** | - | 65.3 | 87.5 |
| halfcheetah-m-r | **71.5** | **78.6** | - | 69.7 | **75.6** | - | 63.4 | 66.2 |
| walker2d-m-r | 108.2 | 103.2 | - | **109.2** | 108.2 | - | 105.4 | **114.3** |
| Mujoco Total | 647.2 | 617.8 | - | 705.5 | 683.4 | - | 698.0 | **753** |

Table 7 shows that PROTO can outperform or obtain comparable performances level compared to **all variants of APL and SUNG** for 8 out of 13 tasks. Furthermore, we can obtain further performance gains via increasing the polyak averaging speed to $10\tau$ (PROTO (UTD=20, $10\tau$)). Although speeding up the updates, the training is stable and we don't observe performance drop, demonstrating PROTO can handle distribution shift in a large variation of hyperparameter changes.

# F   ABLATION STUDY

In this section, we conduct ablation studies on the two hyper-parameters of *PROTO*, the Polyak averaging speed $\tau$ and the conservatism strength annealing speed $\eta$, to investigate whether *PROTO* is hyper-parameter robust.

For $\tau$, we ablates on three sets of parameters: $0.5\tau, 1\tau$ and $2\tau$, where $1\tau$ is the original hyper-parameter that is used to reproduce the results in our paper including $5 \times 10^{-3}$ for all MuJoCo locomotion tasks and $5 \times 10^{-5}$ for all Adroit manipulation and Antmaze navigation tasks. $0.5\tau$ represents half the original speed and $2\tau$ denotes double the original speed. The aggregated learning curves and full results can be found in Figure 12 and Figure 13.

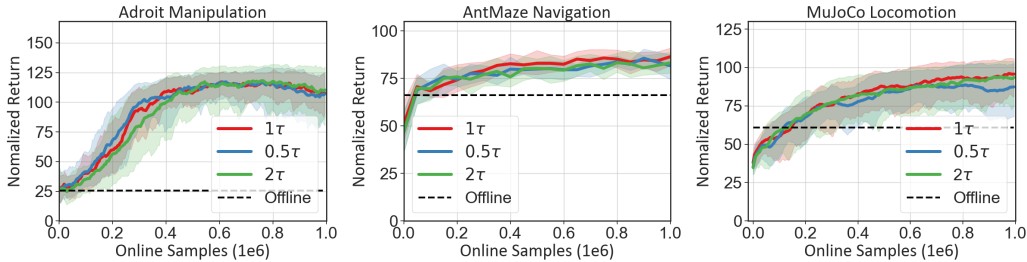

Figure 12: Aggregated learning curves of ablations on the polyak averaging speed.

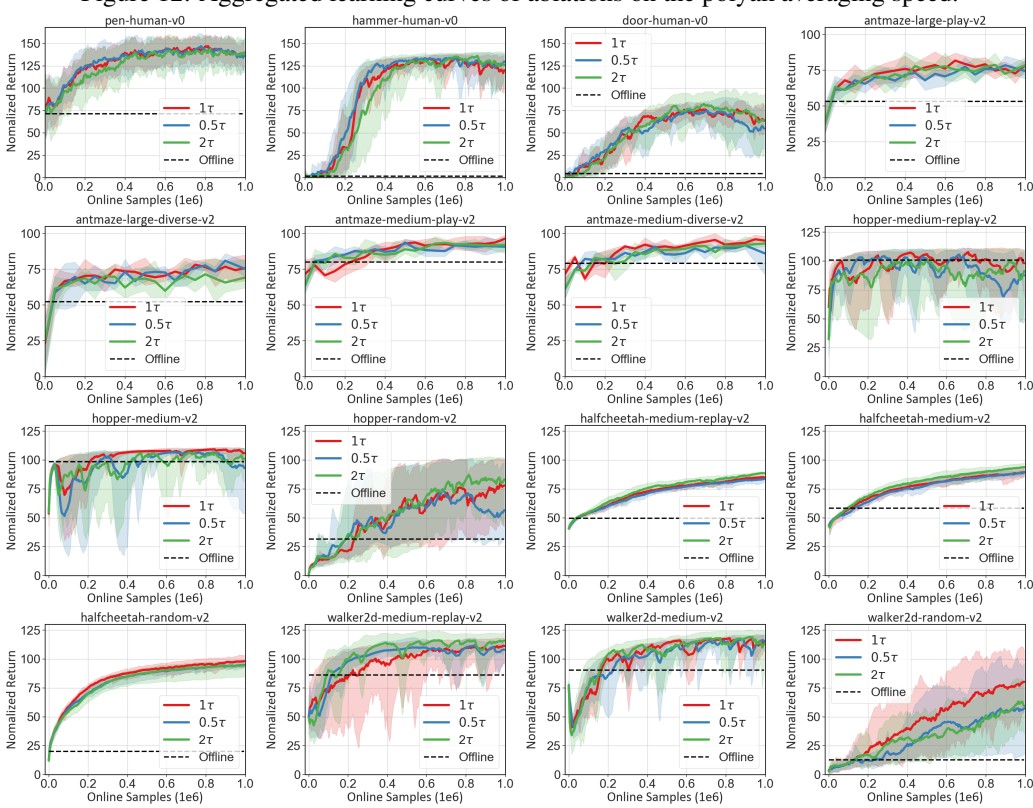

Figure 13: Full results of ablations on the polyak averaging speed.

Figure 12 and Figure 13 illustrate that *PROTO* can achieve similar performances across a range of Polyak averaging speeds, including $0.5\tau$ to $2\tau$. This finding highlights the robustness of *PROTO* to variations in the Polyak averaging speed ($\tau$).

For the ablations on the conservatism annealing speed $\eta$, we also ablate on three sets of parameters: $0.8, 0.9$ and $1.0$, where $0.9$ is the original hyper-parameter that is used to reproduce the results in our paper and means the conservatism strength $\alpha$ anneals to 0 with $\frac{10^6}{0.9}$ online samples. $0.8$ and $1.0$

represents $\alpha$ decays to 0 with $\frac{10^6}{0.8}$ and $\frac{10^6}{1.0}$ online samples, respectively. The aggregated learning curves and full results can be found in Figure 14 and Figure 15.

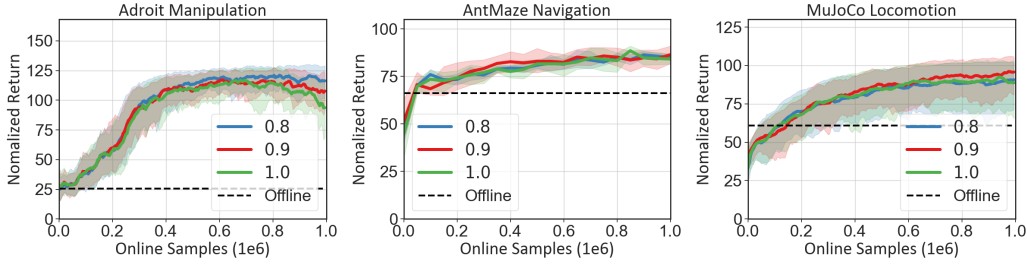

Figure 14: Aggregated learning curves of ablations on the conservatism annealing speed $\eta$.

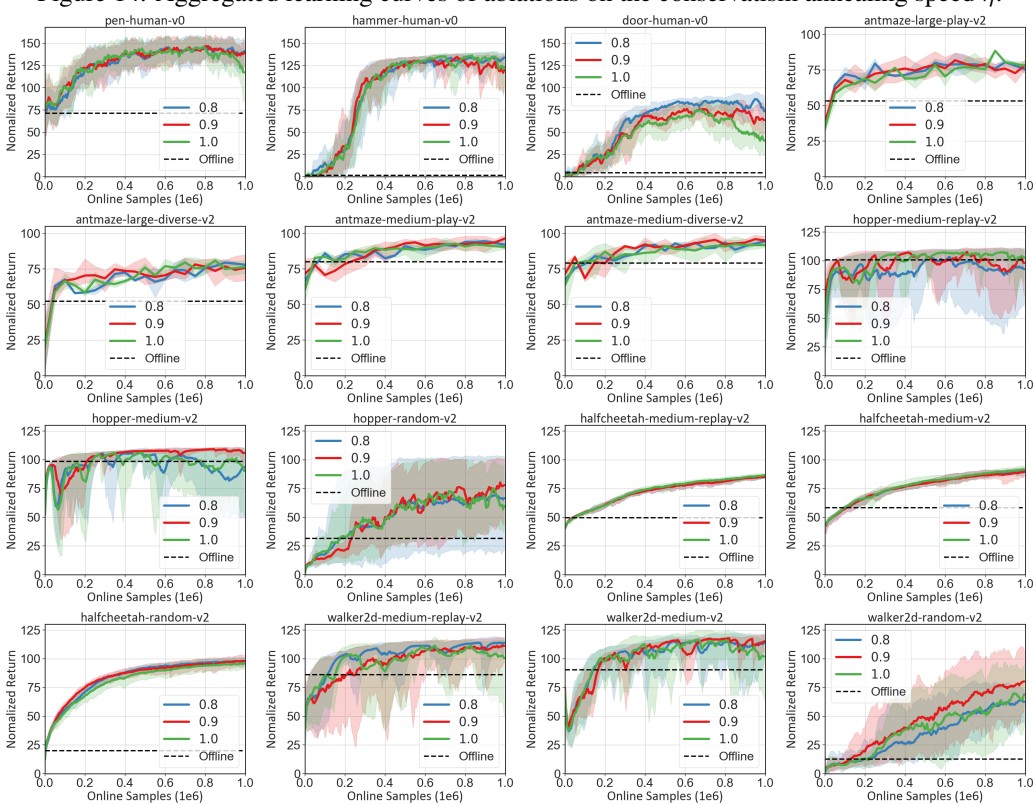

Figure 15: Full results of ablations on the conservatism annealing speed $\eta$.

Figure 14 and Figure 15 demonstrate that *PROTO* can obtain consistently good performances with three sets of annealing speed. In this paper, we adopt a non-parametric treatment by setting $\eta$ as 0.9 to ease the parameter tuning.

## G  FULL RESULTS OF PROTO WITH BC PRETRAINING

In this section, we present the complete results for *PROTO* with BC pretraining, referred to as *PROTO+BC*, to demonstrate the versatility of *PROTO* for various offline pretraining approaches. The results are reported in Figure 16 and Figure 17. In the case of using BC for policy pretraining, we use Fitted Q evaluation (FQE) (Le et al., 2019) to obtain the action-value $Q^0$ that corresponds to the BC policy. FQE is simple to train and insensitive to hyper-parameters, so pretraining $Q^0$ using FQE will not add additional parameter tuning burden or significant computational costs.

It has been observed in prior work (Lee et al., 2022; Nair et al., 2020) that directly initializing the value function with FQE and the policy with BC can lead to a significant performance drop during the initial finetuning stage. However, as shown in Figure 16 and Figure 17, this issue can be effectively addressed while achieving competitive finetuning performances using *PROTO*. This suggests that *PROTO* enables the use of the simplest pretraining method to construct state-of-the-art offline-to-online RL methods, bypassing the need for complex offline RL training.

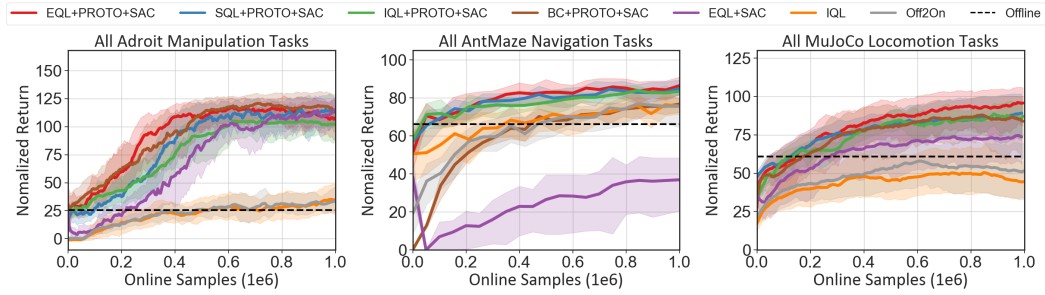

Figure 16: Aggregated learning curves of online finetuning with BC pretrained policy.

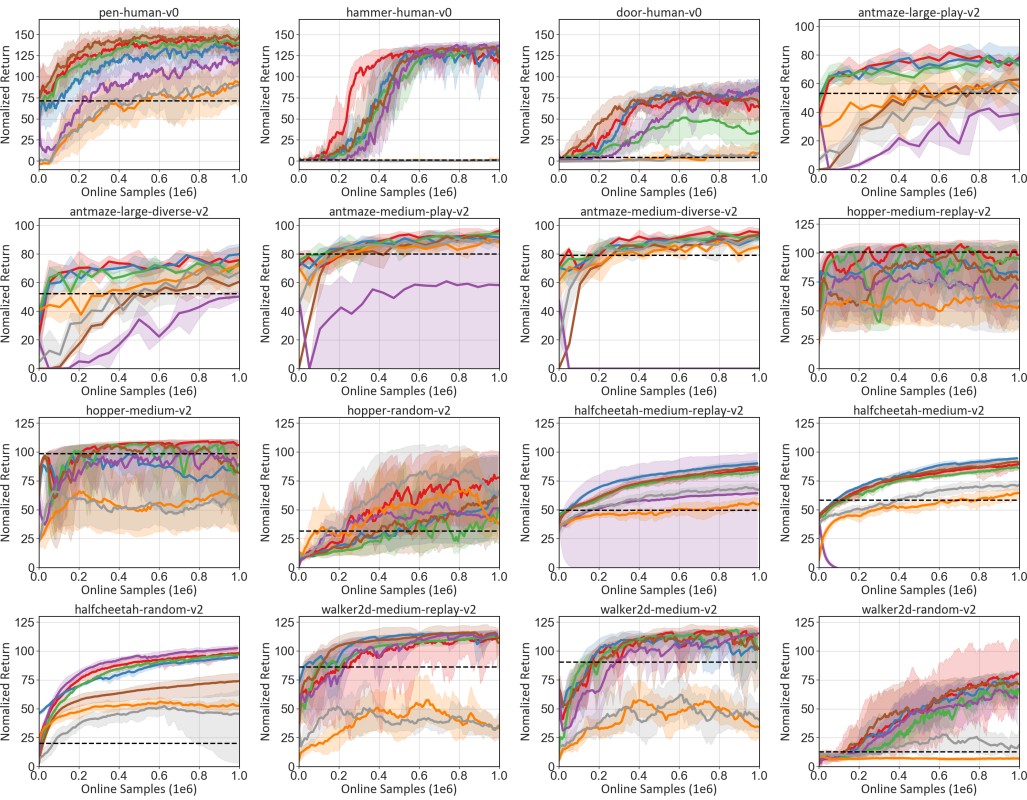

Figure 17: Full results of online finetuning with BC pretrained policy.

## H  FULL RESULTS OF ONLINE FINETUNING WITH TD3

In this section, we plug *PROTO* into TD3 (Fujimoto et al., 2018), another SOTA online RL method that focuses on deterministic policy learning, dubbed as *PROTO+TD3* to demonstrate the adaptability of *PROTO* for diverse online finetuning approaches. Differ from finetuning with SAC (Haarnoja et al., 2018), TD3 builds on top of deterministic policy and thus the log term in Eq. (3) in not calculable. To solve this, we replace the KL-divergence regularization in Eq. (3) with a MSE loss akin to (Fujimoto & Gu, 2021). We also only finetune the mean output of the stochastic policy that is pretrained by EQL and drop the variance head to transfer from stochastic to deterministic policy.

$$\pi_{k+1} \leftarrow \arg \max_{\pi} \mathbb{E} \left[ \sum_{t=0}^{\infty} \gamma^t \left( r(s_t, a_t) - \alpha \cdot (\pi(s_t) - \pi_k(s_t))^2 \right) \right], k \in N, \tag{13}$$

then the policy evaluation operator and policy improvement step become:

$$(\mathcal{T}_{\pi_{k-1}}^{\pi_k} Q)(s,a) := r(s,a) + \gamma \mathbb{E}_{s' \sim \mathcal{P}(\cdot|s,a)} \left[ Q(s', \pi_k(s')) - \alpha \cdot (\pi_k(s') - \pi_{k-1}(s'))^2 \right], k \in N^+, \tag{14}$$

$$\pi_{k+1} \leftarrow \arg \max_{\pi} \left[ Q^{\pi_k}(s, \pi(s)) - \alpha \cdot (\pi(s) - \pi_k(s))^2 \right], k \in N. \tag{15}$$

This objective shares the same philosophy of Eq. (3) that constraining the finetuning policy *w.r.t* an iteratively evolving policy $\pi_k$ instead of a fixed $\pi_0$, which degenerates to a recent work (Luo et al., 2023). However, it is worth mentioning that the constraint strength of the MSE loss in Eq. (15) is far more weak than the log-barrier in Eq. (5). For instance,assume the action space ranges from [-1, 1], then the MSE loss in Eq. (15) is at most 4, which may vanish compared to the large action-value $Q$ during policy improvement Eq. (15), while the log-barrier in Eq. (5) may reach $\infty$. Therefore, we adopt the similar treatment in TD3+BC (Fujimoto & Gu, 2021) that introducing a scalar $\lambda$ during policy improvement to rescale the action value to a comparable scale *w.r.t* the MSE loss to stabilize training:

$$\pi_{k+1} \leftarrow \arg \max_{\pi} \left[ \lambda Q^{\pi_k}(s, \pi(s)) - \alpha \cdot (\pi(s) - \pi_k(s))^2 \right], k \in N, \tag{16}$$

where

$$\lambda = \frac{\beta}{\frac{1}{N} \sum_{(s_i, a_i)} |Q(s_i, a_i)|}, \tag{17}$$

where $N$ is batch size and $\beta$ is a hyper-parameter to control the Q scale. Although introducing one additional parameter to tune, we find that setting $\beta = 4$ can achieve consistently good performance across 16 tasks. For the other parameter choice, we reuse almost all of the hyper-parameters from finetuning with SAC as reported in Appendix D and only disable the clip-double Q trick for adroit Manipulation tasks. We report the aggregated learning curves and full results of *PROTO+TD3* in Figure 18 and Figure 19. We also record the learning curves of training SAC and TD3 from scratch.

Figure 18 and Figure 19 demonstrate the adaptability of *PROTO* for diverse online finetuning approaches. By simply plugging *PROTO* into TD3 (Fujimoto et al., 2018), we can form a competitive offline-to-online RL algorithm *PROTO+TD3*, which also obtains SOTA performances compared to SOTA baselines.

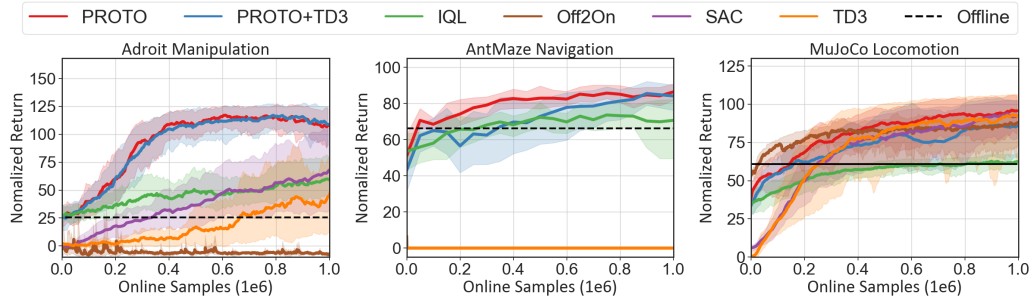

Figure 18: Aggregated learning curves of TD3 online finetuning.

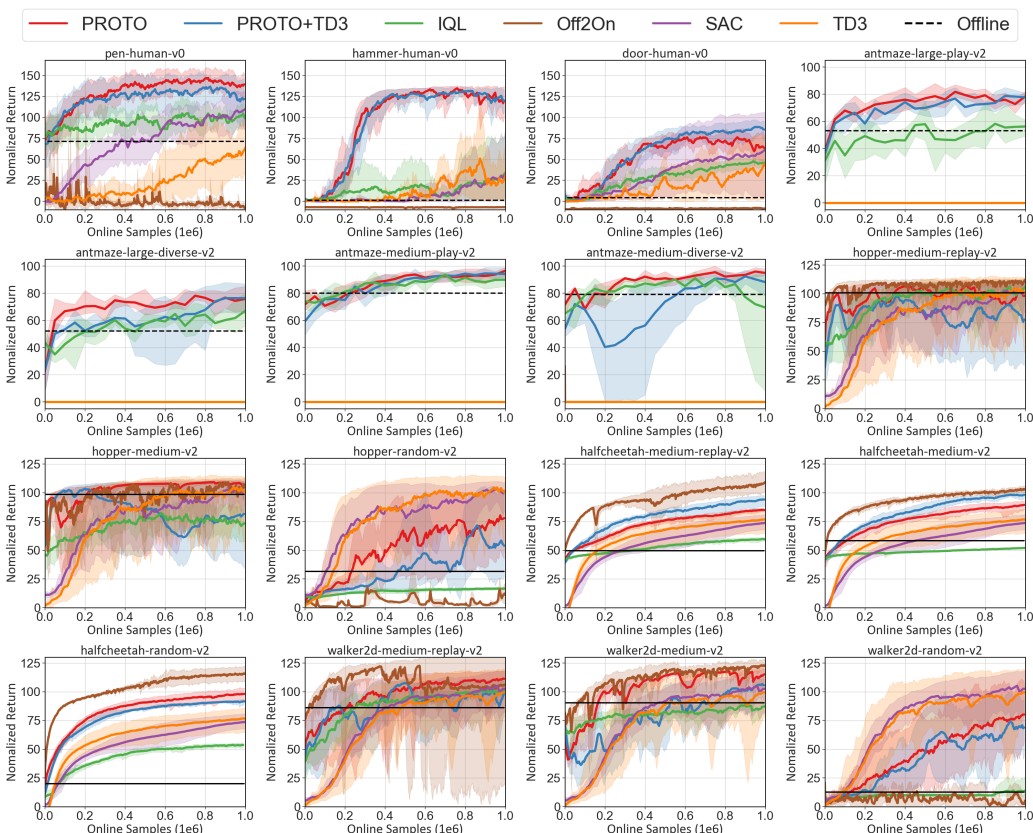

Figure 19: Full results of TD3 online finetuning.

## I  FULL RESULTS OF COMPARISONS WITH FIXED POLICY REGULARIZATION

In this section, we report the aggregated learning curves and full results for the comparisons of *iterative policy regularization* with *fixed policy regularization* in Figure 20 and Figure 21.

Observe from Figure 20 and Figure 21 that although we adopt a linear schedule to anneal the conservatism strength for *Frozen*, *Frozen* still suffers from severe slow online finetuning and poor sample efficiency caused by the initial over-conservatism induced by fixed policy regularization.

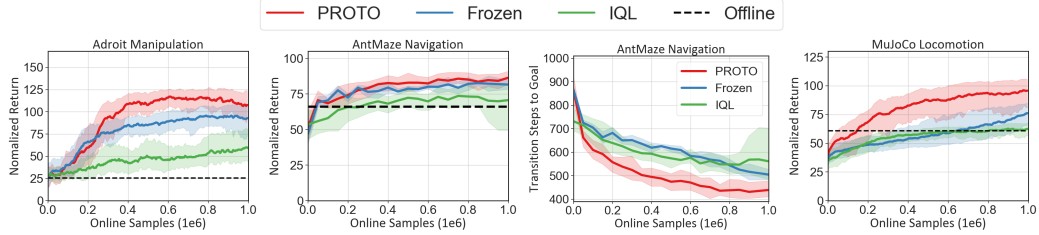

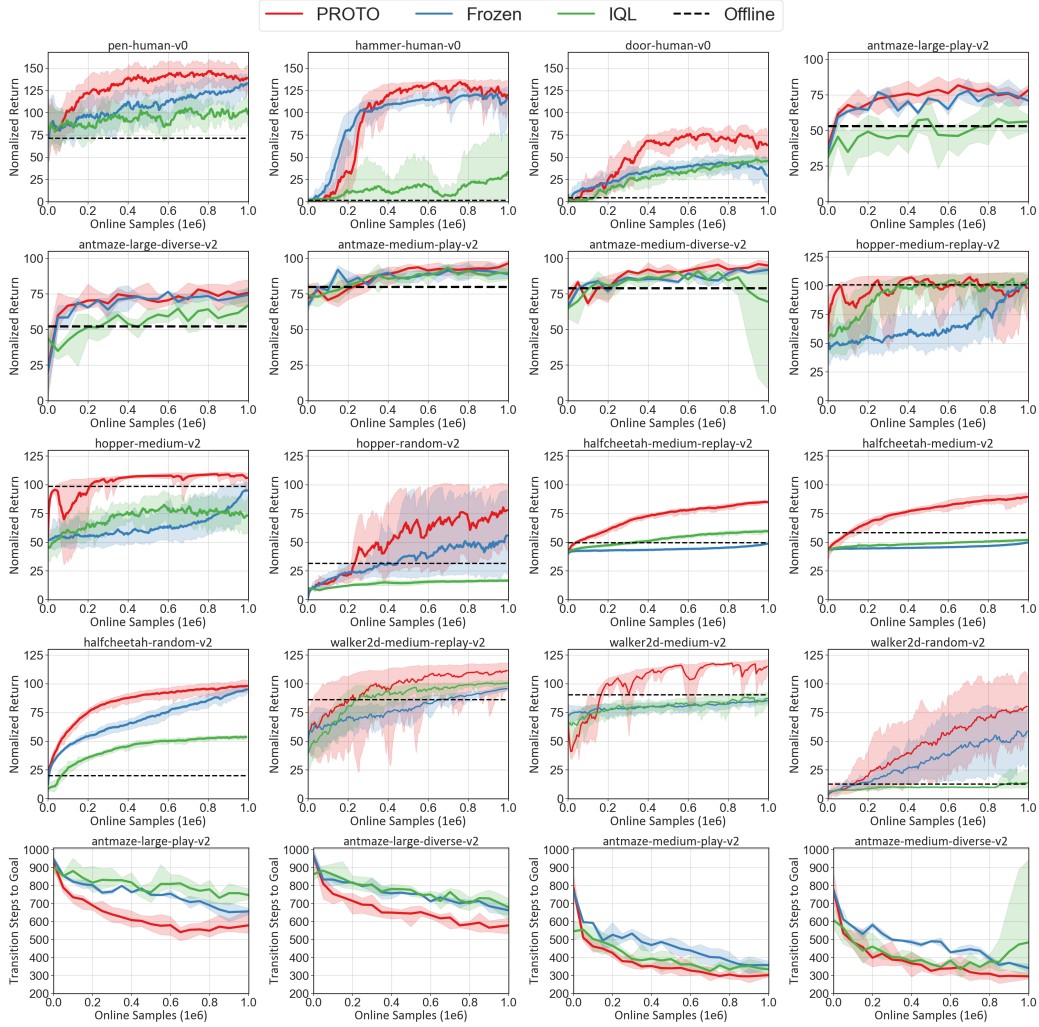

Figure 20: Aggregated learning curves of iterative policy regularization (PROTO) *v.s* fixed policy regularization (Frozen).

Figure 21: Full results of iterative policy regularization (PROTO) *v.s* fixed policy regularization (Frozen).

## J EQL COMPARISON FOR THE OFFLINE2ONLINE SETTING

In this section, we utilize EQL to pretrain and finetune the policy. See from Figure 22 that EQL can achieve stable but suboptimal finetuning performances, as it constrains w.r.t the slowly evolved replay buffer $\mathcal{B}$, which introduces over-conservatism. Moreover, note that the final results of EQL are more suboptimal than IQL. We suspect this is because EQL solves a reverse KL-regularized MDP, which prefers more mode-seeking policies than IQL (Xu et al., 2023). The mode-seeking policy can

benefit the offline pretraining as it avoids OOD issues, but may inhibit the exploration during online finetuning.

Figure 22: EQL comparison for the offline2online setting on Mujoco tasks.

## K    INVESTIGATIONS ON DISTRIBUTIONAL SHIFT DEGREES OF DIFFERENT TYPES OF POLICY REGULARIZATION

In this section, we quantitatively compare iterative, fixed, and no policy regularization to analyze distributional shifts during online finetuning and gain insights into their respective behaviors. Specifically, we record the distributional shift degree w.r.t the pretrained policy $\pi_0$ by measuring the Log-likelihood $\log \pi_0(a|s)$, where $a$ are sampled from the policies that are finetuned by different types of policy regularization methods. The results are presented in Figure 23 (a).

Figure 23 (a) shows that the iterative policy regularization method stays close to $\pi_0$ initially, but gradually deviates from the pretrained policy $\pi_0$ and explores more OOD regions. In contrast, fixed policy regularization always remains close to $\pi_0$ even with annealed policy constraints as it constrains w.r.t $\pi_0$. Also, without any regularization, the finetuned policy quickly shifts away from $\pi_0$, suffering

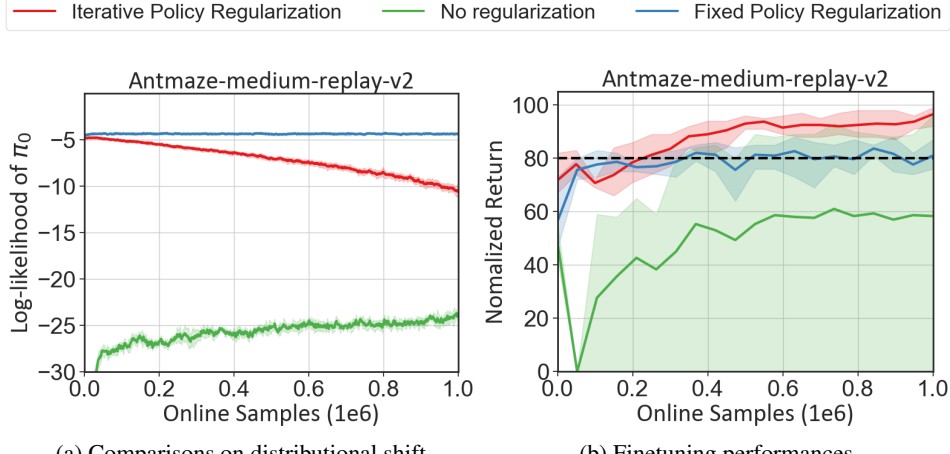

(a) Comparisons on distributional shift      (b) Finetuning performances

Figure 23: Comparisons between iterative/no/fixed policy regularization on distributional shift degrees and finetuning results.

potential instability. Moreover, the results in Figure 23 (b) are consistent with our observations in Figure 23 (a), where no regularization suffers severe instability, fixed regularization undergoes stable but nearly-unchanged results, but iterative regularization enjoys gradually improved performances.

# L   ADDITIONAL ABLATION STUDIES ON THE POLYAK AVERAGING SPEED

In this section, we consider some extreme settings that the polyak average speed $\tau$ is extremely large ($\tau = 0.1$) or no polyak average trick (No $\tau$) is introduced. The results are presented in Figure 24.

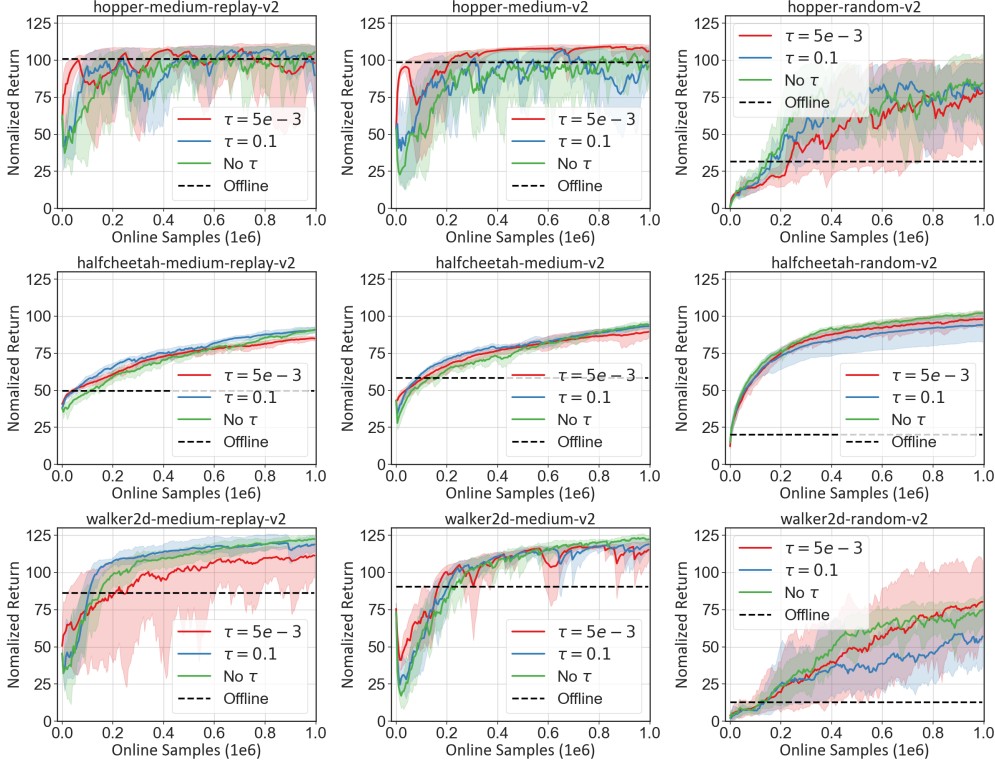

Figure 24: Ablations on the results for no polyak averaging trick. $\tau = 5e - 3$ refers to our default hyper-parameter. No $\tau$ refers to no polyak averaging trick is utilized.

Figure 24 shows that PROTO (No $\tau$) can obtain better results on offline datasets that have good coverage such as medium-replay and random datasets since a good data coverage can alleviate the instability and No $\tau$ can enhance the optimality. However, for datasets that have narrow data coverage such as medium datasets, PROTO (No $\tau$) may undergo some initial performance drop without polyak averaging. Nevertheless, with a stabilized $\bar{\pi}_k$ induced by polyak average, PROTO ($\tau = 5e - 3$) can consistently achieve stable and near-optimal finetuning results, achieving a robust trade-off between stability and optimality across diverse tasks.

We also adopt the polyak averaging trick to directly slow down the EQL+SAC policy updates. Figure 25 shows that without the explicit regularization imposed by KL-regularization, simply adopting the polyak averaging trick into EQL+SAC finetuning cannot still avoid the initial performance drop.

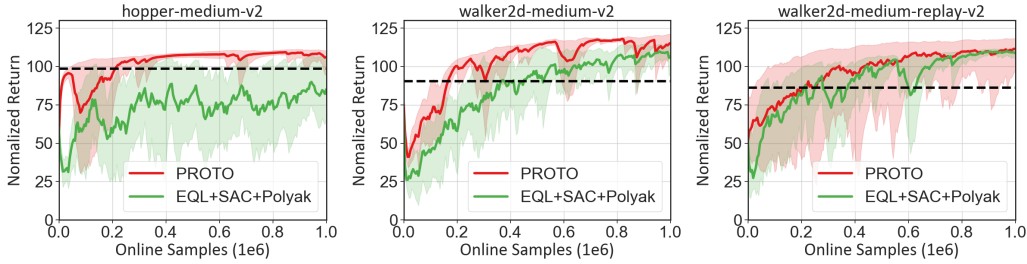

Figure 25: EQL+SAC+Polyak averaging results.

## M    CAL-QL RESULTS

In this section, we compare against a recent SOTA baseline, Cal-QL (Nakamoto et al., 2023). We reproduce the results of Cal-QL using the official codes from `https://github.com/nakamotoo/Cal-QL`. Figure 26 shows that PROTO can outperform or obtain on-par performances compared to Cal-QL on most of the tasks. In addition, Cal-QL is limited to CQL-style pretraining and lacks versatility, while PROTO enjoys great versatility on diverse pretrain and finetune methods.

Note that Cal-QL is not specifically designed for Mujoco tasks and Adroit-human datasets with dense rewards, but tailored for Antmaze tasks with sparse rewards. Therefore, we carefully tune the hyperparameters for Mujoco and Adroit-human tasks to ensure fair comparisons. We observe in the official implementation that Cal-QL introduces a carefully tuned reward scaling+bias trick to enhance performances. However, we find adhering to the original re-scaling scale and bias value cannot lead to reasonable results for Mujoco and Adroit-human tasks. Therefore, we carefully tune these hyperparameters and find setting the reward re-scaling scale as 1 and the bias value as 0 can obtain relatively good results. In this sense, Figure 26 shows that PROTO can still outperform Cal-QL on most of the tasks. We also noticed in a concurrent work (Lei et al., 2023) that Cal-QL cannot achieve satisfactory performances on Mujoco and Adroit-human tasks.

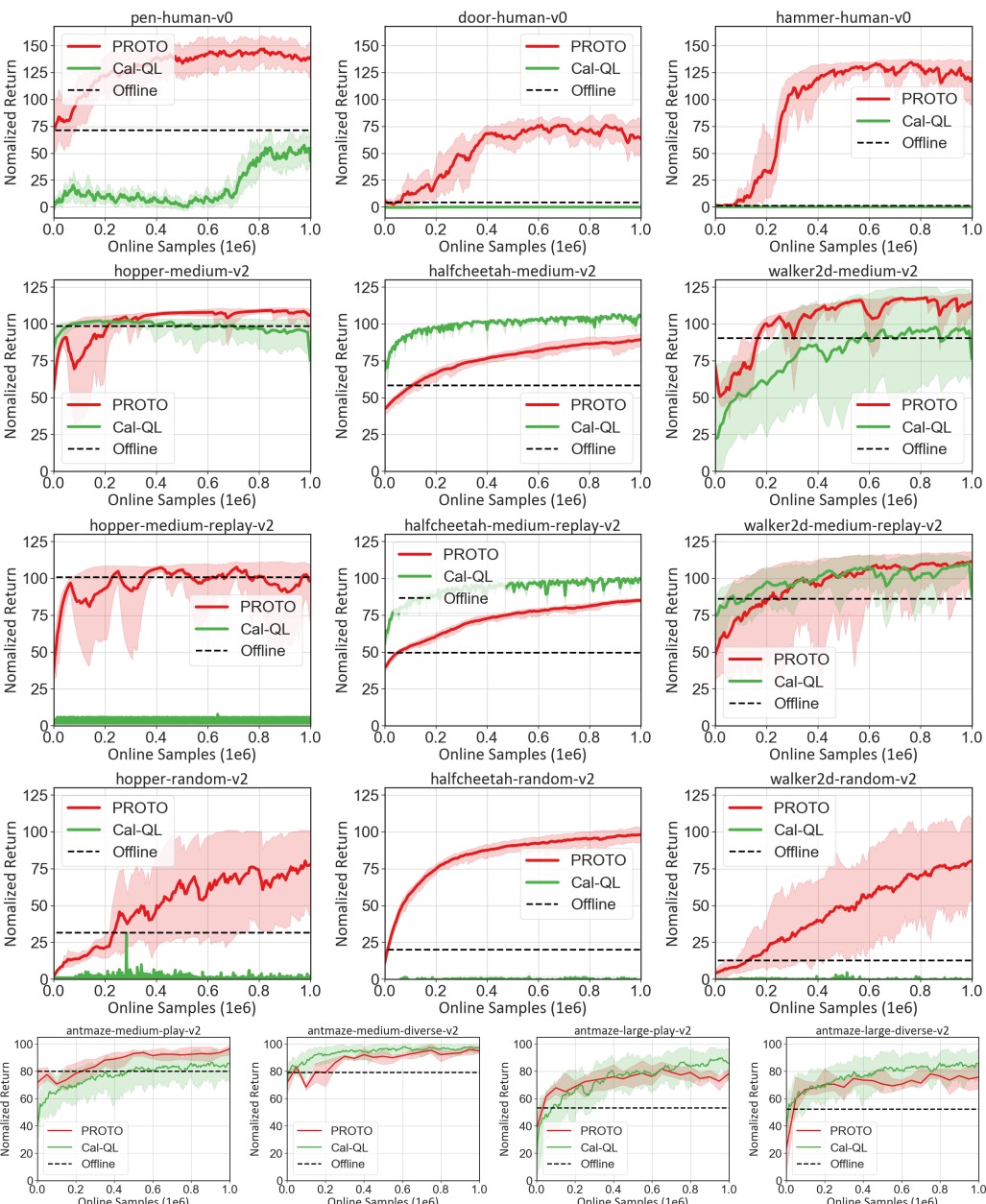

Figure 26: Comparisons with Cal-QL.

