# OpenReview forum: "PROTO: Iterative Policy Regularizied Offline-to-Online Reinforcement Learning"
_ICLR.cc/2024/Conference — Submitted to ICLR 2024_

### Official Review · Reviewer_xy7L · 2023-10-25

**Soundness:** 4 excellent
**Presentation:** 4 excellent
**Contribution:** 3 good
**Rating:** 8
**Confidence:** 4

**Summary:**

This paper proposes a new framework Policy Regularized Offline-To-Online (PROTO) for offline pre-trained to the online fine-tuning RL problem. The proposed PROTO framework can be implemented efficiently, and it also demonstrates improvement upon several existing offline to online methods.

**Strengths:**

1. The paper has conducted a throughout literature review.
2. The limitations of current existing offline-to-online methods are well introduced and explained (Section 3.2).
3. The proposed regularization method is clean and elegant (Equation 3).
4. The intuition of the proposed algorithm is well explained and somewhat theoretically justified (Section 3.3).
5. Most experiments in Section 4 demonstrate the effectiveness of the proposed method.

**Weaknesses:**

1. The experiments in Figure 4 are hard to visualize, perhaps the author can change another color to demonstrate the curves (especially the `BC+PROTO+SAC`, `EQL+SAC`, `PEX+BC`).

**Questions:**

Would it be ok if the author also add the Cal-QL experiments for comparison? As far as the reviewer is aware of, the Cal-QL paper also has open-sourced the [code](https://github.com/nakamotoo/Cal-QL).

---

> ### Author Response · Authors · 2023-11-13
> **Response to Reviewer xy7L**
>
> We are grateful to the reviewer for the constructive comments and positive feedback on our work. Regarding the concerns of the revewer xy7L, we provide the following responses.
>
> >**W1. The experiments in Figure 4 are hard to visualize, perhaps the author can change another color to demonstrate the curves (especially the BC+PROTO+SAC, EQL+SAC, PEX+BC).**
>
> We thank the reviewer for this constructive comment! We have re-drawn almost all figures in our paper according to the suggestions of the reviewer.
>
> >**Q1. Would it be ok if the author also add the Cal-QL experiments for comparison? As far as the reviewer is aware of, the Cal-QL paper also has open-sourced the code.**
>
> We thank the reviewer for this helpful comment. We have added the results in Figure 26 in Appendix M and the hyper-parameter details for Cal-QL in our revised paper. The comparisons show that PROTO can also outperform or obtain on-par performances as compared to Cal-QL.

---

> > ### Comment · Reviewer_xy7L · 2023-11-13
> >
> > Dear authors, thank you very much for the prompt response. I have no further questions and I will maintain my score.

---

> > > ### Author Response · Authors · 2023-11-14
> > > **Thanks for the positive feedback!**
> > >
> > > We really appreciate your positive feedback on our work and your effort engaged in the review process! It's great to know that we've successfully tackled your concerns. Thanks so much for your valuable input!

---

### Official Review · Reviewer_LmwG · 2023-10-29

**Soundness:** 2 fair
**Presentation:** 2 fair
**Contribution:** 2 fair
**Rating:** 3
**Confidence:** 5

**Summary:**

This paper proposes a novel algorithm, PROTO, for offline-to-online RL. After the offline pre-training phase, PROTO introduces the KL entropy regularization term in policy and value function objectives between the current policy and that of the previous iteration, which is iteratively updated, and the coefficient is also linearly decayed to loosen the constraint. PROTO achieves strong performance on adroit manipulation, antmaze navigation, mujoco locomotion from D4RL dataset and exhibits lighter computational cost than prior works.

**Strengths:**

### quality and clarity
- This paper is clearly described and easy to follow.

### significance
- PROTO achieves the notable performance across the benchmark tasks.
- PROTO reduces the computational cost from the existing offline-to-online RL methods such as ODT, Off2On, and PEX, etc.

**Weaknesses:**

- I think this is the naive adaptation of TD3-BC [1] in offline-to-online settings, because KL regularization term results in BC objective in online settings.
- Similar to above, I'm not sure the difference between PROTO and PROTO-TD3 or PROTO-SAC presented in Figure 4/5. I think PROTO and TD3(-BC) is quite similar to each other. Because  PROTO modifies the objective of policy and value function, it is not intuitive to incorporate with SAC.
- Theorem 1 seems the same as what Vieillard et al. (2020) [2] have proven, and equation 7 is the same as what Scherrer et al. (2015) [3] have proven. I don't think there is novel and offline-to-online-specific discussion around the theorem. Please let me know if my understanding is not correct.
- In the similar setting, [4] proposes iterative trust-region update from the offline pre-trained policy.

[1] https://arxiv.org/abs/2106.06860

[2] https://arxiv.org/abs/2003.14089

[3] https://jmlr.org/papers/v16/scherrer15a.html

[4] https://arxiv.org/abs/2006.03647

---
-- Update from the author response --

For the theoretical analysis, I completely disagree with the explanation from the authors. First, the theorem from Vieillard et al., (2020) and Scherrer et al., (2015) only holds under the exact case, where MDP has finite state space and a finite set of actions, and the q-function can be updated from the Bellman equation (this does not mean the update with gradient descent). Because this paper doesn't have any assumptions on those (i.e. PROTO is only proposed with continuous state/action space and function approximation), originally, the discussion does not make any sense. Second, because the theorem from Vieillard et al., (2020) and Scherrer et al., (2015) originally holds under any initialization (initialization of matrix) and thus offline-to-online RL has already been included in online RL, this paper does not have any novel theorem or statement. insightful interpretations for PROTO do not exist. The same statements from Vieillard et al., (2020) and Scherrer et al., (2015) are repeated in the main text. I express strong concerns that the paper has the same theorem as the previous papers without deriving a novel theorem. Moreover, if the important discussion is only discussed in Appendix, that is very weird.

**Questions:**

- Could you clarify the computational cost between SAC and PROTO? Figure 9 may say that PROTO is better than SAC, but I guess there are no algorithmic differences between the two.
- How does PROTO conduct offline pre-training? PROTO also employs iterative policy constraint? or other algorithms (CQL, IQL, etc.)? I guess this algorithm results in TD3-BC [1] in offline settings.
- What is the effect of delayed policy updates (Section 3.4)? Is this necessary?

[1] https://arxiv.org/abs/2106.06860

---

> ### Author Response · Authors · 2023-11-13
> **Clarifications on Two Major Misunderstandings**
>
> We thank the reviewer for the constructive comments. Regarding the concerns of the reviewer LmwG, we provide the following responses.
>
> First, we would like to clarify two major misunderstandings in the Weakness and the Question.
>
> >**W1. I think this is the naive adaptation of TD3-BC in offline2online settings, because KL regularization term results in BC objective in online settings.**
>
> - **No. PROTO is fundamentally different from TD3+BC**.
>   - TD3+BC is over-conservative in offline2online settings. In TD3+BC, the BC regularization only imposes constraints on the slowly evolving data in replay buffer $\mathcal{B}$. However, relying only on $\mathcal{B}$ is too conservative to obtain near-optimal results. To address this, APL[1] and SUNG[2] proposed some adaptive pessimistic methods to reduce the conservatism during online finetuning, but they still cast constraints on the suboptimal $\mathcal{B}$.
>   - In contrast, PROTO considers a gradually evolving trust region induced by $\pi_k$, which is notably superior in offline2online settings. As extensively discussed in our paper, the constraints w.r.t $\pi_k$ enable both stable and near-optimal results. See Figure 1, 2, 3, 6, 7, 8 for comprehensive details.
>   - As thoroughly discussed in Appendix A, the introduction of $\pi_k$ as policy constraints represents a novel and versatile solution that tackles the challenges faced by existing offline2online methods.
>
> >**Q2. How does PROTO conduct offline pre-training? PROTO also employs iterative policy constraint? or other algorithms (CQL, IQL, etc.)? I guess this algorithm results in TD3-BC [1] in offline settings.**
>
> - We must clarify that PROTO is not a specific pretrain/finetune method but a **versatile component** that bridges a wide range of pretrain/finetune methods. It allows us to use almost any policy pretraining methods **without modifications**. Then, we augment actor and critic training in existing off-policy RL methods with the regularization term $\log(\pi/\pi_k)$ to create effective offline-to-online RL methods.
> For example,
>   - EQL+PROTO+SAC involves using EQL for value and policy pretraining and incorporating the iterative policy regularization term $\log(\pi/\pi_k)$ from PROTO into SAC finetuning, bridging the offline2online gap;
>   - SQL+PROTO+SAC uses SQL to pretrain the policy and value networks, then we plug PROTO into SAC for online finetuning;
>   - EQL+PROTO+TD3 refers to using EQL for offline pretraining and then plugging PROTO into TD3 that uses a deterministic policy for online finetuning. Please see Appendix H for EQL+PROTO+TD3 details.
>   - To the best of the authors' knowledge, achieving or evaluating such remarkable versatility has not been reported in prior studies.
> - Also, PROTO and TD3+BC are fundamentally different as discussed above.

---

> ### Author Response · Authors · 2023-11-13
> **Response to concerns**
>
> >**W2. Similar to above, I'm not sure the difference between PROTO and PROTO-TD3 or PROTO-SAC presented in Figure 4/5.**
>
> - Our proposed PROTO is a versatile framework that can bridge a wide range of offline pretraining and online finetuning methods. The default PROTO refers to EQL pretrain+PROTO+SAC finetune as stated in the first paragraph in Section 4 (page 6), and it can be easily adapted to other versions by replacing different online finetuning methods, for example, PROTO+TD3 refers to EQL pretrain+PROTO+TD3 finetune. See Appendix H for PROTO+TD3 details.
>
> >**W3. Theorem 1 seems the same as what Vieillard et al. (2020) [3] have proven, and equation 7 is the same as what Scherrer et al. (2015) [4] have proven. I don't think there is novel and offline-to-online-specific discussion around the theorem. Please let me know if my understanding is not correct.**
>
> - No, there are some differences. Please see Appendix B for the differences between ours and [3][4].
> - Specifically, [3] and [4] only study the conventional online RL settings without considering the offline pretrained policies and value functions, and thus **are not directly applicable in offline2online settings**. However, in our paper, we extend the analysis from [3] and [4] from the conventional online RL to our specific offline2online settings by introducing a mild assumption on $Q_0$ initialization and making several modifications.
> - Note that we aim to provide insightful interpretations for PROTO, as stated in Appendix B, rather than aiming to develop more intricate or stringent bounds.
>
>
> >**W4. BREMEN[5] proposes iterative trust-region update from the offline pre-trained policy.**
>
> We thank the reviewer for providing this related work and we are happy to include this paper in our revision. However, note that BREMEN and PROTO are very different and belong to different problem settings.
> - BREMEN[5] focuses on the deployment-efficient RL setting which involves multiple stages of offline training and online interactions. But PROTO focuses on the offline2online RL setting. These two settings share some similarities but are not the same.
> - BREMEN[5] constrains on a replay buffer $\mathcal{D}$ rather than $\pi_k$ since it periodically resets the policy using the BC policy w.r.t $\mathcal{D}$, see Eq.(3) and Algorithm 1 in [5] for details. The reason why BREMEN also uses trust-region updates is that BREMEN utilizes on-policy RL methods to train the policy using the model rollouts. As the authors stated, BREMEN actually implicitly constrains w.r.t $\mathcal{D}$, as discussed following Eq.(3) in [5]. In contrast, PROTO enforces constraints on $\pi_k$, which has been shown to be superior to the over-conservative replay buffer based constraint.
> - BREMEN adopts a model-based approach and trains an ensemble of dynamics models, which can be computationally expensive. On the other hand, PROTO is model-free and extremely lightweight.
>
>
>
> >**Q1. Figure 9 may say that PROTO is better than SAC**
>
> - No, we did not say PROTO is better than SAC in terms of computational cost in our paper, but "PROTO enjoys the same computational efficiency as SAC since the cost induced by the additional regularization term is negligible".
> - Sorry for the confusion arising from Figure 9, where the bars are too low to visualize the comparisons between SAC and PROTO. Here we translate Figure 9 into the following table for clearer comparisons.
>
> ||ODT|Off2On|PEX|IQL|AWAC|SAC|PROTO|
> |---|---|---|---|---|---|---|---|
> |Time for 1M online steps/minute|1440|480|45|35|30|30|30|
>
> >**Q3. What is the effect of delayed policy updates (Section 3.4)? Is this necessary?**
> - The delayed policy updates serve to enhance stability. Specifically, it provides a smoothly evolving trust region around $\bar\pi$ to regularize the finetuning process.
> - We added Figure 24 in Appendix L in our revised paper to show the role of the polyak averaging trick.
>   - Figure 24 shows that PROTO (No $\tau$) can obtain better results on offline datasets that have good coverage such as medium-replay and random datasets since good data coverage can alleviate the instability and No $\tau$ can enhance the optimality.
>   - However, for datasets that have narrow data coverage such as medium datasets, PROTO (No $\tau$) may undergo some initial performance drop without polyak averaging.
>   - Nevertheless, with a stabilized $\bar\pi_k$ induced by polyak average, PROTO ($\tau=5e-3$) can consistently achieve stable and near-optimal finetuning results, achieving a robust trade-off between stability and optimality across diverse tasks.

---

> > ### Author Response · Authors · 2023-11-13
> > **References**
> >
> > [1] Adaptive Policy Learning for Offline-to-Online Reinforcement Learning. AAAI 2023
> >
> > [2] A Simple Unified Uncertainty-Guided Framework for Offline-to-Online Reinforcement Learning. 2023
> >
> > [3] Leverage the Average: an Analysis of KL Regularization in RL. NeurIPS 2020
> >
> > [4] Approximate Modified Policy Iteration and its Application to the Game of Tetris. JMLR 2015
> >
> > [5] Deployment-Efficient Reinforcement Learning via Model-Based Offline Optimization. ICLR 2020

---

> ### Comment · Reviewer_LmwG · 2023-11-18
>
> Thank you for the detailed response.
>
> My questions on (1) the difference between yours and BREMEN, (2) the discussion on computational cost, and (3) the effect of delayed policy updates become clear now. Here are the remaining comments:
>
> **Comparison to TD3-BC**
>
> I don't think the explanation of the difference between PROTO and TD3-BC is clear. The authors said, `In TD3+BC, the BC regularization only imposes constraints on the slowly evolving data in replay buffer `. What does this mean? I think PROTO also slowly evolves data in the replay buffer since both PROTO and TD3+BC, are off-policy methods. In other words, what is the difference between TD3+BC and PROTO around the replay buffer?  If my understanding is correct, the difference between TD3+BC and PROTO is that TD3+BC considers the KL constraint between the previous policy histories (from replay buffer) and the current policy, while PROTO only considers the policy one-step before and the current policy. This kind of KL constraint (between k-th and k+1-th) has already been proposed in MPO [1] (online RL setting) and adopted in ABM+MPO [2] (offline-to-online RL setting). Even if considering the difference to TD3+BC/MPO,  the algorithmic novelty is still limited.
>
>
> [1] https://arxiv.org/abs/1806.06920
>
> [2] https://arxiv.org/abs/2002.08396
>
>
> **Offline Pretraining**
>
> After reading the author response, I understood the agnostic nature of PROTO for the pertaining. My confusion might come from the lack of explanation about offline pertaining in the methodology section. I only found one sentence in Section 4, which said PROTO uses EQL in this paper. For clarity, it might be better to include one paragraph that mentions (1) PROTO is initialized with EQL in this paper, and (2) PROTO is applicable to any offline RL methods.
>
> **PROTO and PROTO-TD3 or PROTO-SAC**
>
> If my understanding is correct, the difference between PROTO and PROTO-TD3/PROTO-SAC, is whether further finetuning with TD3/SAC is performed. I definitely agree that PROTO can bridge EQL and TD3/SAC in Figure 4/5, but I'm not sure why this bridge is necessary/important. For example, comparing PROTO and PROTO-TD3, PROTO shows faster convergence than PROTO-TD3 in Figure 5. There are no comparisons between PROTO and PROTO-SAC. I think PROTO is already sufficient for offline-to-online RL, and I'm not sure what merit we can enjoy from further SAC/TD3 training. This is a seemingly redundant (unnecessary?) experiment to me.
>
> **Theoretical Analysis**
>
> I completely disagree with the explanation. First, the theorem from Vieillard et al., (2020) and Scherrer et al., (2015) only holds under the exact case, where MDP has finite state space and a finite set of actions, and the q-function can be updated from the Bellman equation (this does not mean the update with gradient descent). Because this paper doesn't have any assumptions on those (i.e. PROTO is only proposed with continuous state/action space and function approximation), originally, the discussion does not make any sense. Second, because the theorem from Vieillard et al., (2020) and Scherrer et al., (2015) originally holds under **any initialization** (initialization of matrix) and thus offline-to-online RL has already been included in online RL, this paper does not have any novel theorem or statement. `insightful interpretations for PROTO` do not exist. The same statements from Vieillard et al., (2020) and Scherrer et al., (2015) are repeated in the main text. I express strong concerns that the paper has the same theorem as the previous papers without deriving a novel theorem. Moreover, if the important discussion is only discussed in Appendix, that is very weird.

---

> ### Author Response · Authors · 2023-11-19
> **Responses to the additional comments of the reviewer LmwG (1/3)**
>
> We appreciate the reviewer for the additional detailed comments. Although the current rating does not meet our expectations, **we find that the reviewer still has some major misunderstandings about our work**. It appears that most of the concerns relate to the novelty and theory contribution. Regarding these remaining concerns and misunderstandings, we provide the following detailed responses.
>
> >**1. Comparison to TD3+BC**
>
> >**1.1. ...TD3+BC considers the KL constraint between the previous policy histories (from replay buffer) and the current policy, while PROTO only considers the policy one-step before and the current policy.**
>
> - Yes, TD3+BC and PROTO are different in terms of the constraints they use, as the reviewer pointed out.
>
> >**1.2 This kind of KL constraint (between k-th and k+1-th) has already been proposed in MPO [1] (online RL setting) and adopted in ABM+MPO [2] (offline-to-online RL setting). Even if considering the difference to TD3+BC/MPO, the algorithmic novelty is still limited.**
>
> **No, we respectfully disagree that our algorithm lacks novelty.**
> - First, the reviewer may have some misunderstandings on ABM+MPO [2].
>   - **ABM+MPO [2] is exclusively an offline RL work that does not consider the offline-to-online setting** (see Algorithm 1 in [2] that no online interactions are involved).
>   - Thus, PROTO and ABM+MPO[2] are fundamentaly different in the problem settings.
> - Second, our main contribution/claim is that we identify that the **trust-region style constraint can inherently address the unique challenges of offline-to-online RL problems**, including the stability-optimality dilemma, limited versatility and computational inefficiency (details in Section 3.2). It is not about claiming to be the first to propose the trust-region update method.
>   - Actually, many methods used trust-region methods to exclusively solve online (e.g., MPO[1]) or offline (e.g., ABM+MPO[2]) RL problems[3-6], as discussed in Appendix A.
>   - This is exactly the motivation of our work that we wonder **whether we can utilize trust-region style updates to bridge the offline-online gap to solve offline-to-online problems**, considering the strong ability of trust-region update, as clearly stated at the beginning of Section 3.3.
>   - Therefore, we kindly believe it is **inappropriate** for the reviewer to assert that PROTO lacks novelty only because some online or offline RL methods have used trust-region updates, since the problem settings are different.
> - Third, the reviewer may think the transfer from constraints w.r.t the replay buffer $\mathcal{B}$ to the last policy $\pi_k$ is simple and trivial. However, we would argue that we demystify a common misconception in prior works that one must require strict constraints or complex design choices to solve offline-to-online RL problems.
>   - **PROTO remains very competitive while providing a much higher degree of simplicity and versatility**, while eliminating a lot of unnecessary complexity, hyperparameter tuning, and computational cost, required by more sophisticated methods.
>   - We believe such simplicity and versatility provide an easy-to-implement baseline for other researchers to construct their competitive offline-to-online RL methods.
>   - Therefore, we believe this simplicity should be considered as **a novelty and a significant contribution**, rather than being mistakenly considered as lacking novelty as the reviewer stated.
> - Last, we provided a thorough literature review of existing offline-to-online RL methods in Table 2 in Appendix A, which has also been acknowledged by other reviewers. The discussions show that **using trust-region style constraint to handle the offline-to-online RL challenges is novel for the offline-to-online RL fields**.
>
> Also, we thank the reviewer for providing these two interesting related works. We have included the discussions on them in Appendix A in our revised paper.

---

> ### Author Response · Authors · 2023-11-19
> **Responses to the additional comments of the reviewer LmwG (2/3)**
>
> >**2. Offline Pretraining**
>
> - Sorry for this confusion. However, note that we also discussed the pretraining and finetuning versatility in Section 3.3 (the Adaptability and Computational Efficiency paragraph). Due to space limit, we only revised some parts in Section 3.3 and Section 4 to make it more clear according to the suggestions. Thanks!
>
> >**3. PROTO and PROTO-TD3 or PROTO-SAC**
>
> >**3.1 For example, comparing PROTO and PROTO-TD3, PROTO shows faster convergence than PROTO-TD3 in Figure 5. There are no comparisons between PROTO and PROTO-SAC.**
>
> - Sorry for the confusion here. In our paper, SAC is the default online finetuning instantiation in our experiment for PROTO, as stated in the first paragraph in Section 4. PROTO shows faster convergence than PROTO+TD3 as SAC adopts an entropy bonus to enhance exploration, but TD3 only utilizes added noise to enhance exploration. We have revised our paper to make this more clear. Thanks!
>
> >**3.2  I think PROTO is already sufficient for offline-to-online RL, and I'm not sure what merit we can enjoy from further SAC/TD3 training. This is a seemingly redundant (unnecessary?) experiment to me.**
>
> - The experiments with SAC/TD3 finetuning demonstrate the versatility of PROTO on diverse off-policy finetuning methods. While PROTO is already sufficient for offline-to-online RL, note that the regularization term $\log(\pi|\pi_k)$ in PROTO is easy to calculate. This good property allows PROTO to act as a pluggable term since the $\log(\pi|\pi_k)$ term can be seamlessly integrated into most off-policy RL methods to mitigate the initial performance drop in offline-to-online settings.
>   -  The experiments confirm this versatility, as both PROTO+SAC and PROTO+TD3 exhibit stable and near-optimal finetuning, while finetuning solely using popular off-policy methods, such as SAC, suffers severe initial performance drop, as shown in Figure 4.

---

> ### Author Response · Authors · 2023-11-19
> **Responses to the additional comments of the reviewer LmwG (3/3)**
>
> >**4. Theoretical Analysis**
>
> >**the theorem from Vieillard et al., (2020) and Scherrer et al., (2015) originally holds under any initialization and thus offline-to-online RL has already been included in online RL, this paper does not have any novel theorem or statement...I express strong concerns that the paper has the same theorem as the previous papers without deriving a novel theorem. Moreover, if the important discussion is only discussed in Appendix, that is very weird.**
>
> The reviewer's primary concern appears to revolve around our theoretical contributions.
> - First, we must clarify that **we never argued to provide some entirely new or even tighter bounds than previous works**, as clearly stated in our paper. We have revised our paper to make it more clear.
> - Instead, we only argued that **we can extend some existing theories from online to our specific offline-to-online setting** to compare different regularizations and bring some theoretical insights. This is clearly stated in the paragraph preceding Eq. (6) in the main text and Appendix B.
>   - We have discussed these previous theories with proper context, citations, and did not claim to introduce entirely novel theoretical contributions in our paper, as we focus more on providing a strong, simple, versatile and efficient offline-to-online framework rather than presenting a theoretical study to provide lots of performance bounds. In this sense, we used existing theories to analyze the proposed methods. To clarify this, we revised our paper to call them Lemma rather than Theorem.
> - Furthermore, we respectfully disagree that our extension to the offline-to-online setting is trivial.
>   - While previous theories assume the $Q_0$ under any initialization, which is versatile and implicitly includes offline-to-online RL settings, we specifically consider such theory in offline-to-online RL settings by introducing one additional Q assumption that the offline pretrained policy should be obtained by one step policy improvement step upon the randomly-initialized $Q_0$. This aspect has not been explored in previous works.
> - Moreover, note that PROTO offers a versatile framework that is capable of bridging diverse offline/online RL methods. In this sense, it becomes pretty **challenging for anyone to provide further detailed theoretical analysis**. Since different instantiation choices will introduce different theoretical properties. To address this, we have empirically verified the versatility of PROTO by extensive experiments in Figure 4, 5.
> - Finally, the discussions in Appendix B primarily aim to explain how we can apply previous theories to our offline-to-online settings. We left this content in the Appendix to avoid overwhelming readers with these detailed explanations and due to the space limits. We provided many links to Appendix B in the main text to guide the readers to read these explanations.
>
> **In summary, we respectfully disagree that the theoretical aspects of our paper are over-claimed.**
>
> >**The theorem from Vieillard et al., (2020) and Scherrer et al., (2015) only holds under the exact case, where MDP has finite state space and a finite set of actions, and the q-function can be updated from the Bellman equation (this does not mean the update with gradient descent). Because this paper doesn't have any assumptions on those (i.e. PROTO is only proposed with continuous state/action space and function approximation), originally, the discussion does not make any sense.**
> - As discussed in Vieillard et al., (2020), they also utilized the theories to analyze the methods in continuous state/action space and function approximation, such as SAC, TRPO and MPO (see Table 1 in Vieillard et al., (2020) for details).
> - We believe that it is reasonable to analyze in the simplified discrete settings to avoid overloading the discussion with terms and assumptions, such as the realizability/completeness assumption, policy class complexity and critic network complexity induced by functional approximation[7]. Also, the simplification from continuous settings with functional approximation to discrete settings has been widely adopted in previous works to facilitate analysis, including but not limited to [8-10].
> - Also note that the focus of our study is to provide a strong practical algorithm rather than presenting a theoretical study to provide lots of performance bounds. Therefore, it may place a heavy burden on the readers to consider the settings with functional approximation that overload the discussion with terms and assumptions.
> - Nevertheless, we appreciate the reviewer's input, and we have made revisions to our paper in the main text and Appendix B to enhance clarity based on these considerations.
>
> **We hope that our additional clarifications are helpful for the reviewer in re-evaluating our paper. We will be more than happy to address any remaining concerns the reviewer may have. Sincerely thank you for your time and looking forward to your response.**

---

> > ### Author Response · Authors · 2023-11-19
> > **Additional References**
> >
> > [1] https://arxiv.org/abs/1806.06920
> >
> > [2] https://arxiv.org/abs/2002.08396
> >
> > [3] Trust region policy optimization. ICML 2015.
> >
> > [4] Proximal policy optimization algorithms. 2017.
> >
> > [5] Behavior proximal policy optimization. ICLR 2023.
> >
> > [6] Trust-PCL: An off-policy trust region method for continuous control. ICLR 2018.
> >
> > [7] Bellman-consistent Pessimism for Offline Reinforcement Learning. NeurIPS 2021.
> >
> > [8] Versatile Offline Imitation from Observations and Examples via
> > Regularized State-Occupancy Matching. ICML 2022.
> >
> > [9] The In-sample Softmax for Offline RL. ICLR 2023.
> >
> > [10] Mildly Conservative Q-Learning for Offline Reinforcement Learning. NeurIPS 2022.

---

### Official Review · Reviewer_6tXk · 2023-11-01

**Soundness:** 4 excellent
**Presentation:** 4 excellent
**Contribution:** 3 good
**Rating:** 6
**Confidence:** 4

**Summary:**

This paper introduces an iterative strategy regularization approach for offline to online reinforcement learning, called PROTO. The method is designed to address three main challenges: poor performance, limited adaptability, and low computational efficiency. By incorporating an iteratively evolved regularization term, the algorithm aims to stabilize the initial online fine-tuning and provide sufficient flexibility for policy learning. PROTO seamlessly bridges various offline reinforcement learning and standard off-policy reinforcement learning, offering a flexible and efficient solution for offline to online reinforcement learning.

**Strengths:**

1. The paper provides a thorough analysis of the three issues present in existing offline to online finetuning methods and proposes a straightforward and versatile solution that effectively overcomes these problems.
2. Furthermore, the article includes rigorous theoretical analysis and extensive experimental validation simultaneously, resulting in a high level of completeness in the paper.

**Weaknesses:**

1. I have some concerns regarding Polyak averaging. Is the initial policy $\bar{\pi}_0$ obtained by the offline pretraining? If so, does Polyak averaging potentially lead to PROTO deviating very slowly from the offline pretrained policy since the parameter is a very small value ($\tau=5e-5,5e-3$)? Additionally, it seems that the role of Polyak averaging might **overlap** with the addition of the KL term in Equation 3. Therefore, I believe that ablation experiments for this parameter are not sufficient. An experiment without this parameter should be conducted. Also, does the update process of EQL+SAC make use of this parameter $\tau$? If not, could this parameter potentially help prevent training collapse in EQL+SAC?
2. Based on the analysis and evidence presented in the paper, I believe that PROTO's capabilities extend beyond improving methods that are already capable of offline to online finetuning. Can PROTO also cooperate with other algorithms that cannot solve the offline to online problem?

**Questions:**

please answer the questions in weaknesses.

---

> ### Author Response · Authors · 2023-11-13
> **Response to Reviewer 6tXk**
>
> We thank the reviewer for the constructive comments and positive feedback on our work. Regarding the concerns of the revewer 6tXk, we provide the following responses.
>
> >**W1.1 Is the initial policy $\pi_0$ obtained by the offline pretraining?**
>
> Yes.
>
> >**W1.2 If so, does a small Polyak averaging (tau=5e−5,5e−3) potentially lead to PROTO deviating very slowly from the offline pretrained policy?**
> - No. Figure 8 shows that PROTO deviates a lot from $\pi_0$ and even obtains a similar deviation degree compared to Off2On, which poses no regularization.
> - Also, we added quantitative investigations on the distributional shift degrees in Figure 23, Appendix K in our revised paper. The results show that PROTO with a very small $\tau$ can still deviate fast from $\pi_0$.
>
> >**W1.3 The role of Polyak averaging might overlap with the addition of the KL term in Equation 3**
> - No. The polyak averaging trick is only utilized to provide a stabilized trust region around $\bar\pi_k$, but is not used to directly slow down the policy updates. In that sense, the polyak averaging assists the KL term by smoothing the potential rapid evolution of the trust region.
> - However, we believe the reviewer makes an excellent idea and insightful observation! Using the polyak averaging trick to directly slow down the policy updates may serve as some kind of implicit policy constraint, while the KL term in Eq.(3) performs as an explicit constraint. Therefore, it would be interesting if we could combine them together to develop more advanced and versatile offline2online RL methods.
>
> >**W2.1 An experiment without polyak averaging should be conducted.**
>
> We thank the reviewer for this helpful comment.
>
> - We added new results for $\tau=0.1$ and no polyak averaging (No $\tau$) in Figure 24, Appendix L in our revised paper.
>   - Figure 24 shows that PROTO (No $\tau$) can obtain better results on offline datasets that have good coverage such as medium-replay and random datasets since a good data coverage can alleviate the instability and No $\tau$ can enhance the optimality.
>   - However, for datasets that have narrow data coverage such as medium datasets, PROTO (No $\tau$) may undergo some initial performance drop without polyak averaging.
>   - Nevertheless, with a stabilized $\bar\pi_k$ induced by polyak average, PROTO ($\tau=5e-3$) can consistently achieve stable and near-optimal finetuning results, achieving a robust trade-off between stability and optimality across all evaluated tasks.
>
> >**W2.2 Does the update process of EQL+SAC make use of $\tau$?**
> - No, the polyak averaging trick is only used to obtain a stable trust region $\bar\pi_k$ to regularize the online finetuning, but not to directly slow down the policy updates. EQL+SAC has no trust region update mechanisms as it directly finetunes without any regularization.
>
> >**W2.3 Could this parameter potentially help prevent training collapse in EQL+SAC?**
>
> - No, we added results for EQL+SAC using polyak averaging with the same $\tau$ as PROTO in Figure 25, Appendix L in our revised paper. The results show that simply using polyak averaging without explicit KL-regularization still suffers training instability in EQL+SAC. Although this attempt fails, we think it is an interesting direction to further explore whether we could utilize the implicit policy regularization induced by polyak averaging to further enhance the results.
>
> >**W3. Can PROTO also cooperate with other algorithms that cannot solve the offline to online problem?**
> - Yes. SAC and TD3 in our paper are online RL methods that cannot directly solve offline2online problems since no regularization is introduced to combat the initial performance drop during online finetuning. Figure 3, 4, 5 show that with PROTO stabilization, all PROTO+SAC and PROTO+TD3 variants obtain stable and near-optimal performances. This demonstrates the versatility of PROTO on diverse online RL methods that cannot directly obtain stable offline2online results.

---

> > ### Comment · Reviewer_6tXk · 2023-11-22
> >
> > The authors provided very detailed responses and addressed each of our concerns thoroughly, adding rich experiments or charts. We maintain our score and recommend to accept this paper.

---

### Official Review · Reviewer_Ykto · 2023-11-10

**Soundness:** 3 good
**Presentation:** 3 good
**Contribution:** 3 good
**Rating:** 6
**Confidence:** 2

**Summary:**

This work propose to conduct KL-regularized policy gradient iterative in the offline-to-online setting. By iteratively update the old policy with the more recent policy and constrain KL distance between current policy and old policy, it achieves a good trade off for stability and optimality.

The authors conducted experiments on D4RL tasks and shows improved performance compared to AWAC, IQL etcs. Additionally, the ablation results show that iteration of policy is essential to the improvement. Also the larger the distance between final policy and initial policy, the better the final performance.

**Strengths:**

- The proposed iterative approach is simple, effective, and novel, as far as I know.
- The ablation results matched with the motivation.
- The experimental results is promising.

**Weaknesses:**

- Are we missing EQL comparison? The EQL should also work with offline to online setting. If we are using EQL as a pre-trained approach, how does the proposed approach perform compared to EQL in the offline to online setting?
- In this setting, the KL-regularized MDP is a moving target, this means the optimal policy per iteration is changing. This seems conflicts with the motivation of the approach. Could the author clarify the intuition here? I assume the \pi* is the optimal policy of original MDP (without KL-constraint).

**Questions:**

- It would be better to clarify the updating frequency of \pi_k controlled by \tau. How sensitive this parameter is when we switching to different tasks and settings? Especially when we have different level of \pi_0. Is there a clear guidance how we should choose \tau?
- How many iterations of policies in total in practice? How many gradient steps we did one delayed updates?
- I think it is good to see that in Figure 7, there is a clear shift of distribution in terms of horizon. It might be more clear to see if there is state-action distribution shift. I am trying to understand if this iterative approach can reach OOD state-action pairs gradually, compared to the \pi_0. I think this is important because: 1) with this verification, we can then understand if the learning can be stable and gradually move to an area deviated from initial state-action coverage. 2) We are not just re-weighting the existing trajectories in the offline dataset.

---

> ### Author Response · Authors · 2023-11-13
> **Response to Reviewer Ykto**
>
> We thank the reviewer for the constructive comments and positive feedback on our work. Regarding the concerns of the reviewer, we provide the following responses.
>
> >**W1. Missed EQL comparison for offline2online setting.**
>
> We thank the reviewer for this constructive comment.
>
> - We added the EQL comparison for the offline2online setting in Figure 22, Appendix J in our revised paper. Results show that EQL cannot reach near-optimal finetuning results as it constrains the finetuned policy w.r.t the slowly evolved relay buffer like IQL does, which is over-conservative. Moreover, EQL performs more conservatively than IQL since EQL solves a reverse KL-regularization, which behaves in a mode-seeking behavior and lacks explorations.
> - We did not compare against EQL in the initial version since both EQL and IQL are similar in-sample learning methods.
>
> >**W2. The KL-regularized MDP is a moving target. This means the optimal policy per iteration is changing.**
>
> - Yes, the optimal policy per iteration is changing in the KL-regularized MDP, and we denote it as $\pi_k$ in our paper, which is different from the optimal policy $\pi^*$ in the original MDP.
> - However, this does not conflict with our motivation as $\pi_k$ will gradually converge to $\pi^*$ as stated in Eq.(6), which enjoys stable and near-optimal convergence compared to fixed regularization (Eq. (8)) and no regularization (Eq. (7)).
>
> >**Q1. How sensitive is the $\tau$ parameter is when we switching to different tasks and settings? Especially when we have different level of $\pi_0$. Is there a clear guidance how we should choose $\tau?$**
>
> - PROTO is not very sensitive with the $\tau$ parameter.
>
>    - PROTO is robust to varying levels of $\pi_0$ while maintaining the same $\tau$ value. In Figure 4, we obtain different qualities of $\pi_0$ using different offline pretraining methods, but keeping $\tau$ constant. The results show that PROTO is robust to different qualities of $\pi_0$ with the same $\tau$.
>    - PROTO is also robust to a wide range of $\tau$ values. Please see Figure 12, 13 in Appendix F for the ablation studies on $\tau$.
>    - PROTO is also robust to different tasks with the same $\tau$ values. We use the same $\tau=5e-3$ for all 9 Mujoco tasks and $\tau=5e-5$ for all 7 Antmaze and Adroit tasks. Please see Table 3 in Appendix D.2 for hyper-parameter details.
>
>
> - For the guidance, $\tau$ can consistently achieve stable and near-optimal results over a broad range of values, such as [1e-2, 2.5e-3] for all Mujoco tasks (Figure 12, 13 and Table 7). It also behaves robustly across various tasks and different levels of $\pi_0$. Therefore, it is easy to select an appropriate $\tau$. Here, we suggest starting with a small $\tau$ to prioritize stable finetuning and avoid risky outcomes, then gradually increasing its value.
>
> >**Q2. How many iterations of policies in total in practice? How many gradient steps we did one delayed updates?**
>
> During finetuning,
> - 1M policy iterations/gradient updates are performed for all PROTO+SAC-based methods, e.g. EQL+PROTO+SAC and BC+PROTO+SAC, the numbers of policy updates, value updates, and online samples for SAC are the same.
> - 0.5M policy iterations/gradient updates are performed for PROTO+TD3 since TD3 performs one policy update per two value updates.
>
>
> >**Q3. It might be more clear to see if there is state-action distribution shift.**
>
> Good point! We thank the reviewer for this constructive comment.
>
> - We added quantitative comparisons on the distributional shift degrees of different policy regularizations in Figure 23, Appendix K in our revised paper. The results show that the iterative policy regularization method stays close to $\pi_0$ initially and can gradually deviate from the pretrained policy $\pi_0$ to explore more OOD regions. In contrast, fixed policy regularization always remains close to $\pi_0$ as it constrains w.r.t $\pi_0$. Also, without any regularization, the finetuned policy quickly shifts away from $\pi_0$, suffering potential instability.

---

> > ### Comment · Reviewer_Ykto · 2023-11-22
> > **Response**
> >
> > Thanks for the prompt and detailed response.

---

### Author Response · Authors · 2023-11-13
**General response**

We thank all the reviewers for the effort engaged in the review phase. Regarding the concerns of the reviewers, we revised our paper (highlighted in blue text color) and summarized the modifications in the following.

1. (For Reviewer Ykto) We added the EQL comparison for the offline2online setting in Figure 22 in Appendix J.
2. (For Reviewer Ykto, 6tXk) We added the investigations on distributional shift degrees of different types of regularization methods in Figure 23 in Appendix K.
3. (For Reviewer 6tXk, LmwG) We added additional ablations on the polyak averaging speed $\tau$ in Figure 24 in Appendix L.
4. (For Reviewer 6tXk) We added EQL+SAC+Polyak results in Figure 25 in Appendix L.
5. (For Reviewer LmwG) We added the discussions on related work BREMEN in Appendix A.
6. (For Reviewer xy7L) We added comparisons with Cal-QL in Figure 26 in Appendix M.
7. (For Reviewer xy7L) We re-colored Figure 4 and re-drawn almost all figures to better visualize the results.

---

### Meta-Review · Area_Chair_fyJY · 2023-12-10

**Metareview:**

In this paper, the authors propose an algorithm, called PROTO, for the offline-to-online RL problem. PROTO starts with a pre-trained policy learned from the offline stage. At each iteration of the online phase, PROTO solves an optimization problem whose objective is the standard RL objective augmented with an iterative regularization term (a KL term -- negative log of the current policy divide by the one from the previous iteration). The authors provide some theoretical results to support their approach that none is new and it is not clear how they should be interpreted. Then, they slightly modify PROTO and discuss its practical implementation. Finally, they empirically compare it with several algorithms in Adroit manipulation, AntMaze navigation, and MuJoCo locomotion tasks from D4RL.

(+) The paper is easy to read. The algorithm is relatively simple, achieves good performance across the benchmark tasks, and has less computational cost than the existing offline-to-online RL algorithms (e.g., ODT, Off2On, and PEX).

(-) I am puzzled why the authors claim that PROTO is an offline-online RL algorithm. The only role of the offline phase is to initialize the policy for the online stage (pre-training). There is no discussion on the algorithm used in the offline phase. In fact the authors say "... allows us to use almost any policy pre-training method ...". If we can arbitrarily initialize the policy, PROTO can be potentially used for online RL.
(-) PROTO is just doing mirror descent with KL as the Bregman divergence. This is very similar to the on-policy MDPO algorithm in

Mirror Descent Policy Optimization by Tamar et al. (ICLR-2022)

Even to the adaptive and fixed KL algorithms in the original PPO paper. Another reference is

Adaptive trust region policy optimization: Global convergence and faster rates for regularized MDPs by Shani et al. (AAAI-2020)

(-) There is absolutely no novelty in the theoretical results of the paper. Everything on Page 5 has been borrowed from other papers. It is not clear how the results are supposed to provide insightful interpretation for PROTO. The main part of the paper is the experiments section. See the comments by Review LmwG.
(-) There are several inaccurate statements in the paper. For example, in the abstract the authors claim "PROTO yields ... optimal final performance". What is the notion of optimality here?

**Justification For Why Not Higher Score:**

(-) I am puzzled why the authors claim that PROTO is an offline-online RL algorithm. The only role of the offline phase is to initialize the policy for the online stage (pre-training). There is no discussion on the algorithm used in the offline phase. In fact the authors say "... allows us to use almost any policy pre-training method ...". If we can arbitrarily initialize the policy, PROTO can be potentially used for online RL.

(-) PROTO is just doing mirror descent with KL as the Bregman divergence. This is very similar to the on-policy MDPO algorithm in

Mirror Descent Policy Optimization by Tamar et al. (ICLR-2022)

Even to the adaptive and fixed KL algorithms in the original PPO paper. Another reference is

Adaptive trust region policy optimization: Global convergence and faster rates for regularized MDPs by Shani et al. (AAAI-2020)

(-) There is absolutely no novelty in the theoretical results of the paper. Everything on Page 5 has been borrowed from other papers. It is not clear how the results are supposed to provide insightful interpretation for PROTO. The main part of the paper is the experiments section. See the comments by Review LmwG.

(-) There are several inaccurate statements in the paper. For example, in the abstract the authors claim "PROTO yields ... optimal final performance". What is the notion of optimality here?

**Justification For Why Not Lower Score:**

N/A

---

### Decision · Program_Chairs · 2024-01-16

Reject